# Two-dimensional gersiloxenes with tunable bandgap for photocatalytic $H_2$ evolution and $CO_2$ photoreduction to CO

Fulai Zhao [1], Yiyu Feng[1,2 ✉], Yu Wang[1], Xin Zhang[1], Xuejing Liang[1], Zhen Li[1], Fei Zhang[1], Tuo Wang[3,4], Jinlong Gong [3,4] & Wei Feng [1,2,4 ✉]

The discovery of graphene and graphene-like two-dimensional materials has brought fresh vitality to the field of photocatalysis. Bandgap engineering has always been an effective way to make semiconductors more suitable for specific applications such as photocatalysis and optoelectronics. Achieving control over the bandgap helps to improve the light absorption capacity of the semiconductor materials, thereby improving the photocatalytic performance. This work reports two-dimensional −H/−OH terminal-substituted siligenes (gersiloxenes) with tunable bandgap. All gersiloxenes are direct-gap semiconductors and have wide range of light absorption and suitable band positions for light driven water reduction into $H_2$, and $CO_2$ reduction to CO under mild conditions. The gersiloxene with the best performance can provide a maximum CO production of $6.91 \, mmol \, g^{-1} \, h^{-1}$, and a high apparent quantum efficiency (AQE) of 5.95% at 420 nm. This work may open up new insights into the discovery, research and application of new two-dimensional materials in photocatalysis.

[1] School of Materials Science and Engineering, Tianjin University, Tianjin Key Laboratory of Composite and Functional Materials, Tianjin 300072, P. R. China. [2] Key Laboratory of Advanced Ceramics and Machining Technology, Ministry of Education, Tianjin 300072, P. R. China. [3] Key Laboratory for Green Chemical Technology of Ministry of Education, School of Chemical Engineering and Technology, Tianjin University, Tianjin 300072, China. [4] Collaborative Innovation Center of Chemical Science and Engineering, Tianjin 300072, P. R. China. ✉email: fengyiyu@tju.edu.cn; weifeng@tju.edu.cn

Photocatalysis has attracted wide attention due to the high efficiency, low energy consumption, clean, and non-secondary pollution advantages. 2D nanomaterials provide a wide range of opportunities for constructing diverse forms of composite photocatalysts with high activity for photocatalysis, due to their extraordinary advantages such as the atomic thickness, larger surface-to-volume ratio, good conductivity, superior electron mobility, and the high fraction of coordinated unsaturated surface sites[1,2]. However, the important properties that qualifies a 2D crystal for photocatalytic application are the suitable bandgap, band edge levels, optical absorption, and charge carrier mobility[3]. Most 2D materials have several structural limitations as photocatalysts, for example, graphene is zero-bandgap material[4], which is not sufficient to absorb light to drive photocatalytic oxidation or reduction reaction; the absorption range of g-$C_3N_4$ is mainly limited in the ultraviolet region[2]; the monolayer of transition-metal chalcogenides represented by $MoS_2$ and $WS_2$ is direct-bandgap semiconductor, while the bilayer and multilayer are indirect semiconductors[5], which will affect the energy conversion efficiency of light. In addition, most 2D semiconductor photocatalysts need a noble-metal cocatalyst to improve the photocatalytic efficiency[6].

Silicene and germanene are group-IV 2D-Xenes analog to graphene, and are also the so-called zero-bandgap materials but with direct bandgap of 1.55 and 23.9 meV[7,8], respectively. They have better tunability of the bandgap than graphene. Therefore, the bandgap engineering of silicene and germanene has been widely explored in various applications including electronic devices[9,10], photodetectors[11], chemical sensors[12], batteries[13,14], catalysis[15], and topological insulators[16,17]. It is proved by theoretical and experimental results that silicene, germanene and their derivatives have great potential in photocatalysis[18–20], and one of the silicene derivatives has been demonstrated to be a metal-free semiconductor for photocatalytic water splitting[21].

Hydrogenation and alloying are two effective ways to tailor the bandgap[22,23]. The hydrogenation of silicene and germanene have been achieved by the topochemical transformation of Zintl-phase $CaGe_2$ and $CaSi_2$ into germanane (GeH)[24] and silicane (SiH)[25]. Another hydrogenation product of silicene is siloxene ($Si_6H_3(OH)_3$) which has also been synthesized by the similar methods[26]. 2D honeycomb $Si_{1-x}Ge_x$ (siligene) has been demonstrated in theory to be energetically stable because Si and Ge atoms have similar covalent radii, which enable their honeycomb geometry to deform a little to accommodate different atoms[27,28]. Their electronic properties can be tuned by the value of $x$[29]. The $Si_xGe_{1-x}H$[30] alloys are predicted to have finite gaps in the range of 1.09–2.29 eV for $0 \leq x \leq 1$. Ge and Si are known to be completely miscible in any ratio, and GeSi random alloys have been investigated for many years[31–33]. However, the creation of siligenes and their derivatives have rarely been reported. The synthesis of 2D Ge/Si alloy analogues of germanane and silicane would allow a better understanding of how the electronic structure, optical properties can be tuned to realize enhanced optoelectronic properties and photocatalysis applications.

In the study, we report the freestanding siligenes terminated with −H/−OH ($Ge_{1-x}Si_xH_{1-y}(OH)_y$, $x = 0.1 - 0.9$) and name them gersiloxenes, which are synthesized by the topochemical transformation of freestanding $Ca(Ge_{1-x}Si_x)_2$ alloys prepared by annealing stoichiometric ratios of calcium, germanium and silicon. By combining the experimental results with theoretical calculations, we demonstrate their direct gap type and the bandgap dependence on $x$ (the content of Si), which increase with $x$ values from 1.8 to 2.57 eV. The as-synthesized gersiloxene with $x = 0.5$ (HGeSiOH) is most suitable for photocatalytic hydrogen production and reduction of $CO_2$ to CO than the siloxene, germanane, or other gersiloxene samples due to its moderate band edge levels and bandgap, hybridized orbital composition of the valence band (VB)

and conduction band (CB), wide spectral response range, high specific surface area, and oxygen vacancies in gersiloxenes. This gersiloxene generates $H_2$ at a rate of 1.58 mmol $g^{-1}$ $h^{-1}$ in photocatalytic water reduction and CO as the product at a rate of 6.91 mmol $g^{-1}$ $h^{-1}$ in $CO_2$ photoreduction under mild conditions (25 °C, 1 atm $CO_2$) and without additional noble-metal cocatalysts.

## Results

**Characterization of the resulting materials.** The synthesis of gersiloxenes was accomplished by the typical topotactic deintercalation of the Zintl-phase precursor $CaGe_{2-2x}Si_{2x}$ ($x = 0.1$, 0.3, 0.5, 0.7, and 0.9). Crystals of $CaGe_{2-2x}Si_{2x}$ were synthesized by sealing stoichiometric amounts of Ca, Ge, and Si inside a quartz tube at temperature of 1000–1200 °C. The as-prepared $CaGe_{2-2x}Si_{2x}$ crystals are bright black crystals with metallic luster and have stacking lamellar microstructure (Supplementary Figs. 1, 2, details in Supplementary Note 1). X-ray diffraction (XRD) patterns (Supplementary Fig. 3) demonstrated they have the same trigonal rhombohedral tr6 crystal structure as that of $CaSi_2$ and $CaGe_2$[34]. The lattice constant a gradually changes from 3.9837 to 3.8613 Å as $x$ increases from 0.1 to 0.9 (Supplementary Table 1), following the Vegard's law (details in Supplementary Note 1). These Zintl-phase $CaGe_{2-2x}Si_{2x}$ crystals were converted into 2D $Ge_{1-x}Si_xH_{1-y}(OH)_y$ ($x = 0.1 - 0.9$) by topotactic deintercalation in aqueous HCl at −30 °C for 3–10 days. Based on subsequent structural characterization, we present a schematic diagram of the transformation, as shown in Fig. 1a. With $x < 0.5$, the conversion of 2D $CaGe_{2-2x}Si_{2x}$ crystals to $Ge_{1-x}Si_xH_{1-y}(OH)_y$ is similar to that of $CaGe_{2-2x}Sn_{2x}$ crystals into 2D $Ge_{1-x}Sn_xH_{1-x}(OH)_x$ ($x = 0 - 0.09$)[35] because the Si−O bond (800 kJ $mol^{-1}$) is much stronger than the Ge−O (660 kJ $mol^{-1}$), Ge−H (320 kJ $mol^{-1}$), and Si−H bonds (300 kJ $mol^{-1}$);[36] moreover, Ge−O bonds can be readily cleaved in hydrochloric acid[37], thereby forming 2D GeSi alloys terminated with Ge−H and Si−OH that are connected to honeycomb GeSi planes, i.e., $(GeH)_{1-x}(SiOH)_x$. When $x \geq 0.5$, Si−H bonds appeared; therefore, every Ge atom is terminated with −H, and Si atoms are terminated with either −H or −OH above or below the layer, i.e., forming $(GeH)_{1-x}Si_x(OH)_{0.5}H_{x-0.5}$. The overall topochemical transformation reactions are as follows[38].

$$CaGe_{2-2x}Si_{2x} + 2HCl + xH_2O \rightarrow 2(GeH)_{1-x}(SiOH)_x \\ + CaCl_2 + xH_2, x < 0.5 \tag{1}$$

$$CaGe_{2-2x}Si_{2x} + 2HCl + H_2O \rightarrow 2(GeH)_{1-x}Si_x(OH)_{0.5}H_{x-0.5} \\ + CaCl_2 + H_2, x \geq 0.5 \tag{2}$$

XRD patterns of the as-prepared products (Fig. 1b) obtained from the topotactic deintercalation of $CaGe_{2-2x}Si_{2x}$ showed a diffraction structure similar to that of GeH[19,24] and $Si_6H_3(OH)_3$[39] and no phase separation of the $Ge_{1-x}Si_xH_{1-y}(OH)_y$ alloys into the pure phases GeH and $Si_6H_3(OH)_3$. Samples with $x = 0.1$ and 0.3 showed the 6R phase typical of GeH, while for $x = 0.5$, 0.7, and 0.9, the samples exhibited the same 1 T phase as that of $Si_6H_3(OH)_3$. The major diffraction peaks at ~15–16° and 26–28° were indexed to the (006) and (012) planes (for $x = 0.1$ and 0.3) or (001) and (100) (for $x = 0.5$, 0.7, and 0.9)[40], respectively. It can be seen that with increasing $x$ values, the angle of $2\theta$ corresponding to the (006) or (001) crystal plane decreases gradually, while the corresponding interplanar spacing increases gradually. This phenomenon was caused by the increase in Si content together with the −OH termination, which was conducive to larger interlayer spacing. The angle of $2\theta$ corresponding to the (012) or

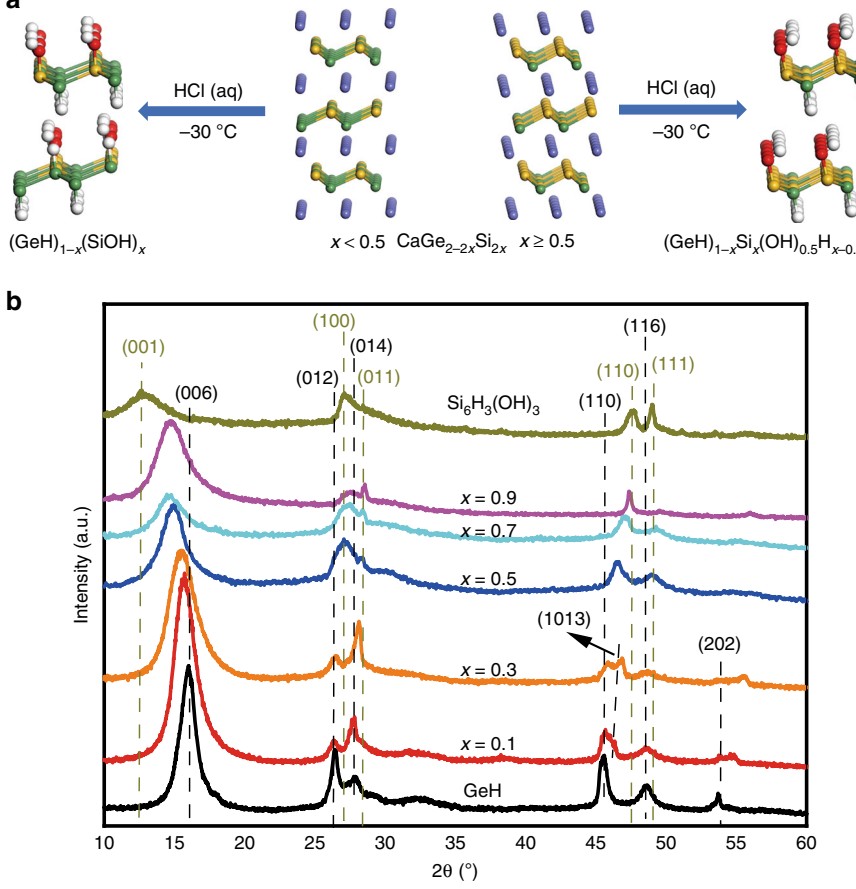

**Fig. 1 Crystal structures of CaGe$_{2-2x}$Si$_{2x}$ and gersiloxenes. a** Schematic illustration of topotactic deintercalation of CaGe$_{2-2x}$Si$_{2x}$ to gersiloxenes (GeH)$_{1-x}$ (SiOH)$_x$ ($x < 0.5$) or (GeH)$_{1-x}$Si$_x$(OH)$_{0.5}$H$_{x-0.5}$ ($x \geq 0.5$) (Ca, blue; Ge, green; H, white; O, red; Si, yellow). **b** XRD patterns of the topotactic deintercalation products gersiloxenes with varying $x$ ($x = 0.1, 0.3, 0.5, 0.7, 0.9$), GeH and Si$_6$H$_3$(OH)$_3$. XRD, X-ray diffraction. Source data are provided as a Source Data file.

(100) crystal plane changed little with the increase in $x$, which indicated interplanar distances of 3.324, 3.346, 3.286, 3.287, and 3.243 Å for $x = 0.1 - 0.9$, respectively. This characteristic can be further confirmed by transmission electron microscopy (TEM).

Figure 2 showed TEM images of as-prepared Ge$_{1-x}$Si$_x$H$_{1-y}$ (OH)$_y$ nanosheets. All the samples had a graphene-like layered morphology with wrinkles similar to that reported for GeH[19] and Si$_6$H$_3$(OH)$_3$[41]. It is also apparent that the number of layers in the samples with $x = 0.5$ and $x = 0.7$ is significantly less than that in the other samples because their transparency and wrinkles are significantly greater. High-resolution transmission electron microscopy (HRTEM) images showed clear lattice fringes, and selected area electron diffraction (SAED) patterns of samples exhibited two sets of hexagonally arranged diffraction spots, revealing the high-quality crystalline structure of 2D Ge$_{1-x}$Si$_x$H$_{1-y}$(OH)$_y$. From the HRTEM images, we can observe that the crystal spacings of d$_{(012)x=0.1}$, d$_{(012)x=0.3}$, d$_{(100)x=0.5}$, d$_{(100)x=0.7}$, d$_{(100)x=0.9}$ were 0.334, 0.330, 0.322, 0.326, and 0.318 nm, respectively, which match well with the powder XRD patterns. Atomic force microscope (AFM) was used to confirm the several atomic layer thickness of 2D gersiloxenes. As shown in Fig. 3, the thickness of the nanosheets was measured to be ~3–6 nm.

Fourier transform infrared (FTIR) spectroscopy was used to confirm the chemical structure of the as-prepared 2D gersiloxenes. For comparison, GeH showed extremely strong Ge−H stretching at ~2000 cm$^{-1}$ and multiple wagging modes at 570 and 479 cm$^{-1}$. The weak doublet peak at 780 and 845 cm$^{-1}$ is due to GeH$_2$

bending modes from neighboring Ge atoms at the edges of each crystalline germanane sheet[24,42]. For Si$_6$H$_3$(OH)$_3$, the bands observed at 519, 640, 876, 1056, 1638, 2116, and 3406 cm$^{-1}$ correspond to the vibrations of ν(Si−Si), δ(Si−H), ν(Si−OH), ν (Si−O), ν(OH), ν(Si−H), ν((Si3)≡Si−H), and ν(OH)[43,44], respectively. It can be seen from the FTIR spectrum (Fig. 4a) of 2D gersiloxenes (with $x$ values from 0.1 to 0.9) that the vibration peaks of Ge−H stretching and Si−OH stretching always exist regardless of the value of $x$. With increasing $x$ value, the Ge−H bond shifted to high wavenumbers. This shift is caused by Si−OH groups, as the Si−OH group exhibits a stronger dipole moment than the Si−H group, thus affecting the stretching vibration of Ge−H groups on adjacent germanium atoms[43]. Moreover, the vibration peak of Si−H stretching only appears when $x > 0.5$, and the intensity ratio of the Si−H to Ge−H vibration peaks increases with the $x$ value. The Si−Si stretching vibration peak also becomes more prominent than that of the Ge−H wag when $x >$ 0.5. Raman spectroscopy was employed to further study the chemical structure of 2D gersiloxenes. As shown in Fig. 4b, peaks at 281–292 cm$^{-1}$, 484–495 cm$^{-1}$, and 388–413 cm$^{-1}$ can be attributed to the phonon modes of Ge−Ge, Si−Si and Ge−Si bonds[44,45], respectively. Peaks at ~640 and ~730 cm$^{-1}$ can be assigned to Si−H bonds[46], and broad peaks at ~2100 cm$^{-1}$ are assigned to Si−H$_2$, which is present only at the sides of silicon nanosheets[46,47]. It is known that crystalline Ge and Si exhibit Raman peaks at 300 and 520 cm$^{-1}$ that correspond to Ge−Ge bonds and Si−Si bonds, respectively. Compared to those of the corresponding bonds in crystalline germanium and silicon, the Raman peaks of both the

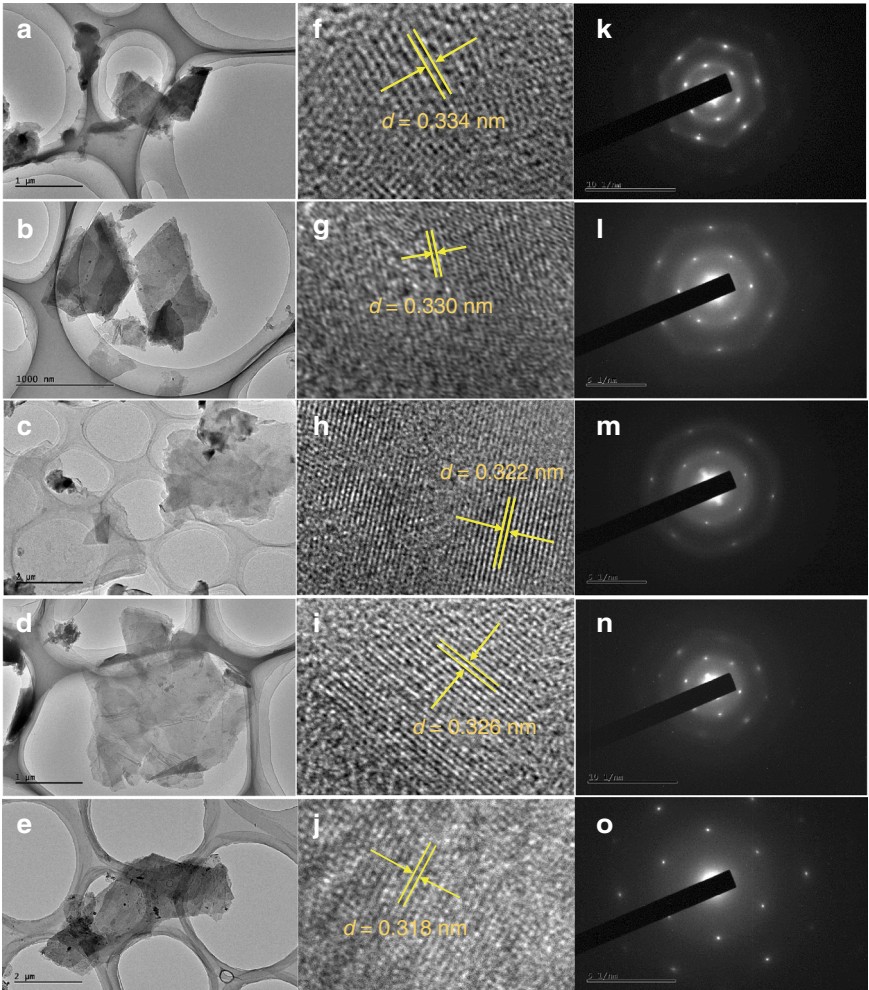

**Fig. 2 Morphology and structural characterization of gersiloxenes. a–e** Low-magnification TEM images, **f–j** HRTEM micrograph, and **k–o** electron diffraction patterns of gersiloxenes sheets. **a, f, k,** $x = 0.1$; **b, g, l,** $x = 0.3$; **c, h, m,** $x = 0.5$; **d, i, n,** $x = 0.7$; **e, j, o,** $x = 0.9$. TEM transmission electron microscopy, HRTEM high-resolution transmission electron microscopy.

Ge−Ge and Si−Si bonds in the gersiloxenes showed a redshift. This result is similar to that for GeSi alloy nanocrystals[32] and 2D germanane and silicane[24,48], which is attributed to the quantum size confinement effects in two-dimensional nanoscale materials[33,49]. It is worth noting that the vibrational mode of Si−H and Si−H$_2$ peaks appears only when $x > 0.5$, indicating that when $x \leq 0.5$, only hydroxyl groups are bonded to Si, while when $x > 0.5$, either hydrogen atoms or hydroxyl groups are bonded to Si. This finding is consistent with the FTIR results. Moreover, with the increase in $x$ value, both the Ge−Ge and Si−Si peaks broadened, indicating that the crystallinity of the 2D gersiloxenes exhibits a slight decrease with increasing $x$ value. It is noteworthy that the Si−Si bond was not detected in the sample with $x = 0.1$ and that the Ge−Ge bond was not detected in the sample with $x = 0.9$. This result may be due to the uniform distribution of GeSi atoms in the two systems, which separate Si from Si atoms by Ge atoms (for $x = 0.1$) or Ge from Ge atoms (for $x = 0.9$) by Ge−Si bonds. It may also be that the proportion of Si−Si or Ge−Ge bonds in the structure is too small to be detected. X-ray photoelectron spectroscopy (XPS) was performed to confirm the configuration of the as-prepared 2D gersiloxenes. The XPS survey spectra showed that all 2D Ge$_{1-x}$Si$_x$H$_{1-y}$(OH)$_y$ samples contain elements of Ge, Si, O, and Cl (Fig. 4c). The high-resolution Ge3d XPS spectra exhibit peaks at ca. 30 eV, corresponding to the Ge−Ge bonds (Fig. 4d), and the shoulder peaks around 32.7 eV are ascribed to Ge−O bonds, which

is due to the trace oxidation on the surface similar to that of germanium nanosheets[50,51]. In the Si2p spectra, the peaks are located at binding energies of approximately 99.9 and 103.0 eV, corresponding to Si−Si bonds and Si−O bonds in the two-dimensional Si chain network (Fig. 4e), respectively. By contrast, the Ge−Ge and Si−Si bonds for the precursor CaGe$_{2-2x}$Si$_{2x}$ are located at 28.8 and 98.5 eV (Supplementary Fig. 4), respectively. This result indicates that CaGe$_{2-2x}$Si$_{2x}$ transforms to 2D Ge$_{1-x}$Si$_x$H$_{1-y}$(OH)$_y$. In addition, the Si2p spectra show that the content of the Si−Si bond relative to that of the Si−O bond in Ge$_{1-x}$Si$_x$H$_{1-y}$(OH)$_y$ increases with an increasing $x$ (when $x = 0.9$, the content of the Si−Si bond is obviously higher than that of the Si−O bond), further indicating that when $x > 0.5$, Si atoms are terminated not only by OH but also by H. Moreover, O1s spectrum in the range of 527–536 eV exhibits the states of surface oxygen on samples (Fig. 4f). The O1s peaks can be also fitted into several components. Peaks at around 531, 532.4, and 533.1 eV are attributed to the Ge−O, Si−O and O−H, respectively. For GeH, Si$_6$H$_3$(OH)$_3$ and all gersiloxenes, a same fitted peak located at around 531.7 eV, corresponded to the adsorbed O atoms in the vicinity of oxygen vacancies (Oads), is supposed to be associated with the surface oxygen vacancies[52,53], which would significantly influence their optical properties.

Scanning electron microscopy (SEM) images (Supplementary Fig. 5) showed that all gersiloxenes had a layered stacking

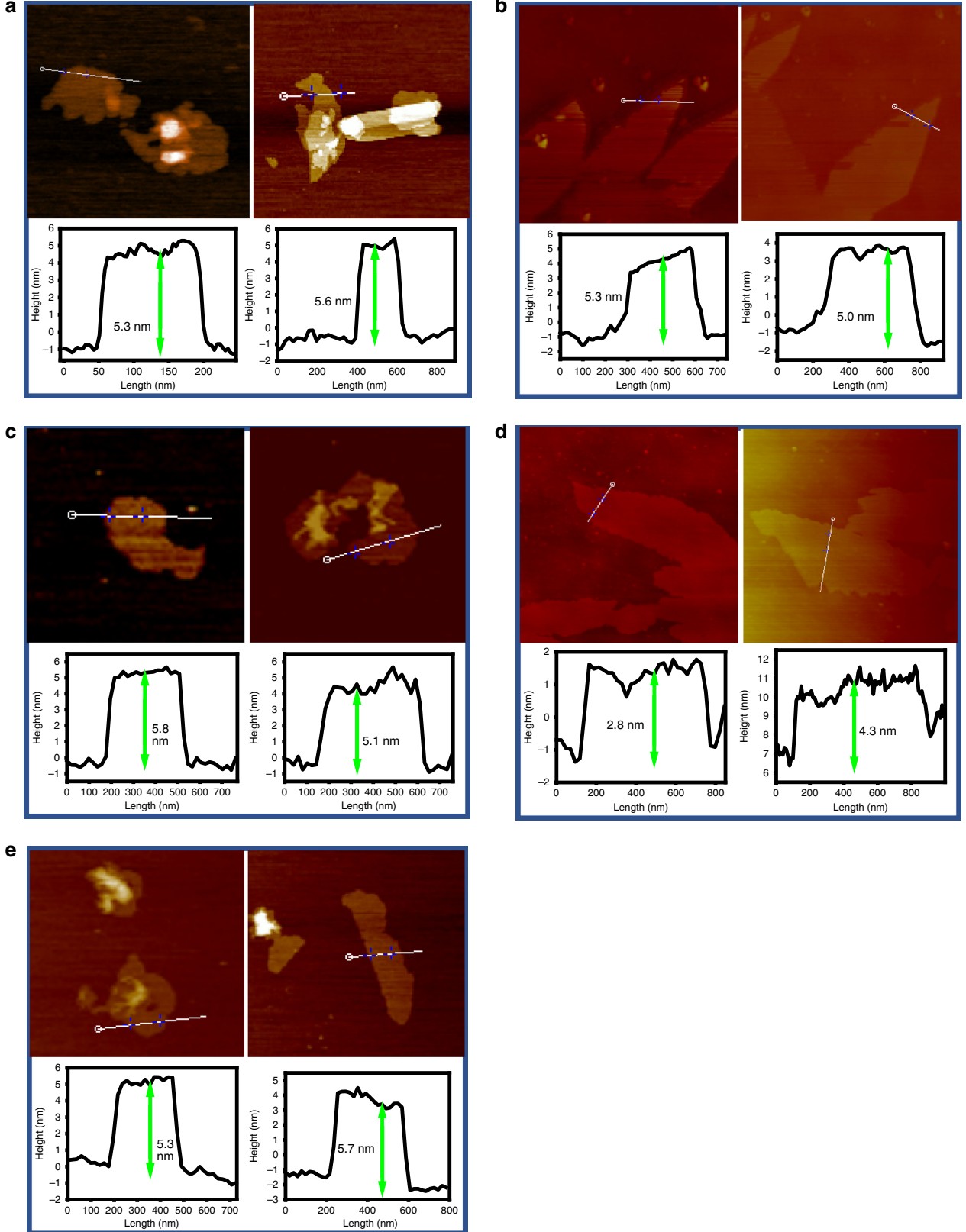

**Fig. 3 Thickness characterization of gersiloxenes.** AFM images and corresponding height profiles of gersiloxenes nanosheets. **a** $x = 0.1$. **b** $x = 0.3$. **c** $x = 0.5$. **d** $x = 0.7$. **e** $x = 0.9$. AFM atomic force microscope. Source data are provided as a Source Data file.

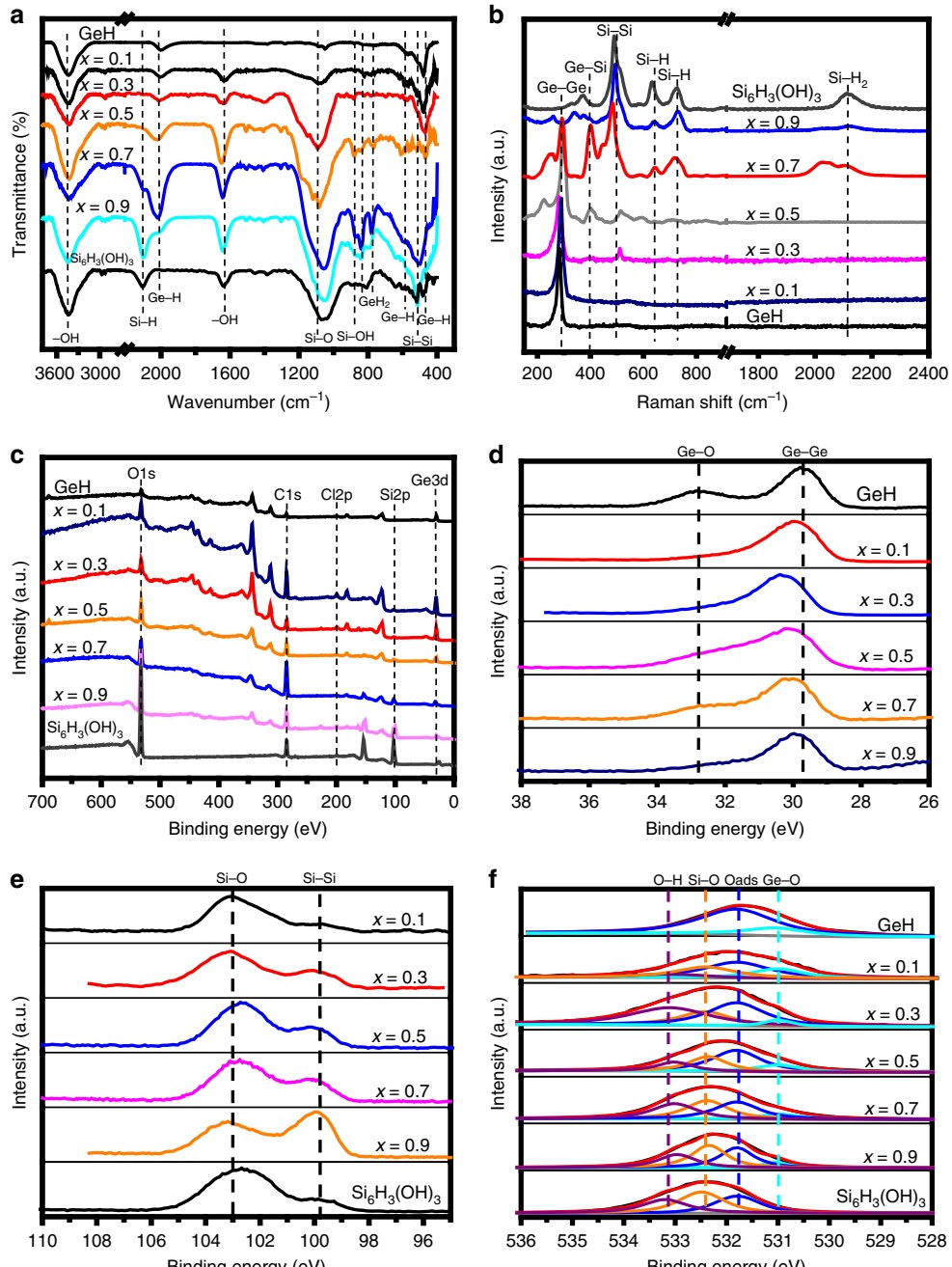

**Fig. 4 Chemical structural characterization of gersiloxenes. a** FTIR and **b** Raman spectra of gersiloxenes with different *x* values, GeH and $Si_6H_3(OH)_3$.
**c–f** XPS spectra of gersiloxenes (*x* = 0.1, 0.3, 0.5, 0.7, 0.9), GeH and $Si_6H_3(OH)_3$. **c** XPS survey spectra. High-resolution XPS spectra of **d** Ge3d **e** Si2p, and
**f** O1s. Oads represents the adsorbed O at vacancy sites. FTIR Fourier transform infrared, XPS X-ray photoelectron spectroscopy. Source data are provided
as a Source Data file.

structure. Gersiloxenes with *x* = 0.1 and 0.3 had regular layered stacking structures with larger lamellar sizes. When *x* = 0.5, the size of the nanosheets decreases significantly, and the lamellae become fragmented. The gersiloxenes with *x* = 0.1 and 0.3 are regular and flat layered structures. However, as *x* increases to *x* = 0.5, the size of the lamella decreases gradually. When *x* > 0.5, the size of the lamellae increases again. In addition, the structure of the gersiloxenes with *x* > 0.5 become distorted and strip or ribbon-like. As shown in Supplementary Fig. 6, SEM elemental mapping of all samples revealed a uniform distribution of Si, Ge and O. The corresponding energy-dispersive spectroscopy (EDS) results helped confirm that the Ge/Si ratios (shown in Table 1) in

the as-synthesized 2D gersiloxenes are almost consistent with the intended *x* values. To investigate the influence of surface area, Brunauer–Emmett–Teller (BET) surface area measurements were carried out. The $N_2$ Adsorption/desorption isotherms and pore-size distribution curves (Supplementary Fig. 7a–d) indicate that all gersiloxenes, GeH, and $Si_6H_3(OH)_3$ have mesoporous structure and gersiloxene with *x* = 0.5 exhibits the most extensive size distribution (details in Supplementary Note 2). Consistent with the differences between SEM images, the surface area continuously increases from 18.9 to 319.7 $m^2 g^{-1}$ for the *x* = 0.1 to *x* = 0.5 samples and then drops to 169.4 $m^2 g^{-1}$ as the x value increases to 0.9 (Table 1). In contrast, GeH and $Si_6H_3(OH)_3$ have

specific surface areas of 4.7 and 94.8 m$^2$ g$^{-1}$, respectively (Supplementary Fig. 7b). The aforementioned TEM results showed that the transparency and wrinkles of gersiloxene nanosheets with $x = 0.5$ is the most obvious of all gersiloxenes, indicating that the nanosheets with $x = 0.5$ is the best dispersed, and the interlayer agglomeration effect is the weakest among all

samples. Moreover, SEM also showed that the overall size of gersiloxene nanosheets with $x = 0.5$ is significantly smaller than that of other gersiloxenes. These factors lead to the largest specific surface area, which can maximize its contact with the liquid[54]. It is reported that the large specific surface area may be beneficial to improve the performance of photocatalysts, as it may provide a larger active area that offer paths for the migration of photogenerated carriers, promote the migration of photogenerated electrons and holes between layers, inhibit the recombination of carriers, and boost the release of generated gases[55].

Characterization by XRD, FTIR, Raman, XPS, SEM, BET, and TEM confirmed the chemical structural characteristics of the synthesized 2D honeycomb GeSi alloys and their physical properties of morphology and surface area. The optical properties of these 2D gersiloxenes were investigated by UV–vis diffuse reflectance spectroscopy (DRS). As shown in Fig. 5a, all gersiloxene samples have a wide range of light absorption from

**Table 1 Summary of the Ge/Si ratios and BET surface areas of 2D gersiloxenes with varying $x$ values.**

| $x$ | 0.1 | 0.3 | 0.5 | 0.7 | 0.9 |
|---|---|---|---|---|---|
| Theoretical Ge/Si ratio | 9:1 | 7:3 | 1:1 | 3:7 | 1:9 |
| Experimental Ge/Si ratio detected by EDS | 8.44:1 | 8.89:3 | 1:1.07 | 3:6.36 | 1:8.77 |
| BET surface Area (m$^2$ g$^{-1}$) | 18.9 | 46.2 | 319.7 | 228.7 | 169.4 |

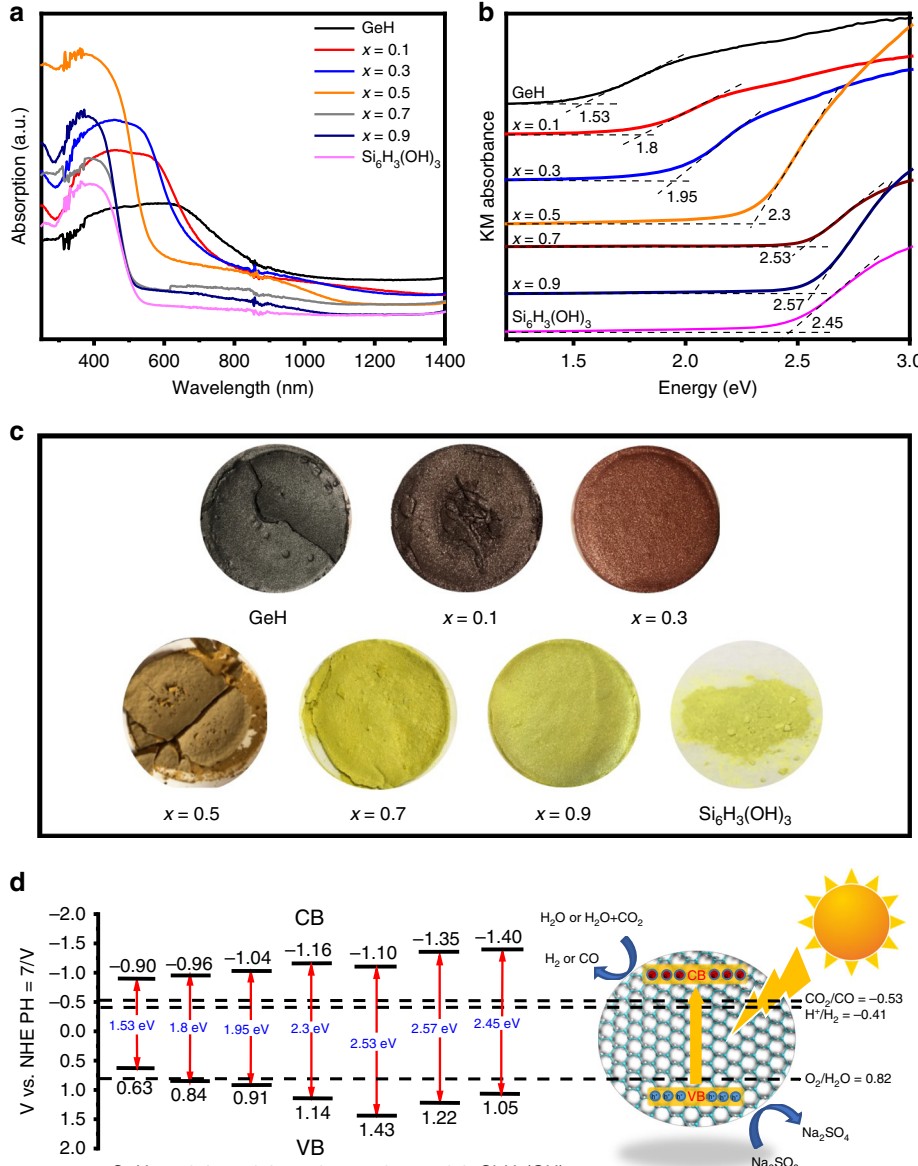

**Fig. 5 Optical properties and energy band structure of gersiloxenes. a** UV–vis diffuse reflectance spectra and **b** Tauc plots of gersiloxenes ($x = 0.1$, 0.3, 0.5, 0.7, 0.9), GeH and Si$_6$H$_3$(OH)$_3$. **c** The optical images of GeH, Si$_6$H$_3$(OH)$_3$, and gersiloxenes with $x = 0.1$, 0.3, 0.5, 0.7, 0.9. **d** Energy band structure of gersiloxenes with different $x$ values and GeH and Si$_6$H$_3$(OH)$_3$ for CO$_2$ reduction to CO and H$_2$ evolution. VB valence band, CB conduction band. Source data are provided as a Source Data file.

ultraviolet to visible light. The absorption range of gersiloxenes is between that of GeH and $Si_6H_3(OH)_3$. With the increasing $x$, the absorption edge shifts to the direction of short wavelength, which is consistent with the change in sample's color from dark red, brick red, brown, yellow green, to light green (Fig. 5c). It is worth noting that gersiloxene shows obvious sub-bandgap absorptions compared with pure GeH and pure $Si_6H_3(OH)_3$. This sub-bandgap absorption is supposed to be caused by the oxygen vacancies, which is indicated by the XPS O1s spectra. Gersiloxene with $x = 0.5$ exhibits the most significant sub-bandgap absorption, indicating that it has the highest oxygen vacancies concentration than other ones which may be attributed to its graphene-like 2D structure and the highest specific surface area among the as-synthesized gersiloxenes. The significant wide sub-bandgap absorption further increases the light absorption range of the material, which is advantageous for the enhanced photocatalytic and photoelectrochemical performance. Figure 5b presents the corresponding Tauc plots, which are calculated based on the assumption that the gersiloxenes are direct-bandgap semiconductor materials and proved by subsequent theoretical calculations. The bandgap values of GeH and $Si_6H_3(OH)_3$ are 1.53 and 2.45 eV, respectively, which is almost in agreement with the reports. For gersiloxenes, with an increasing $x$, the bandgap increased from 1.8 eV ($x = 0.1$) to 2.57 eV ($x = 0.9$), indicating that different $x$ values (the content of Si) endowed the 2D gersiloxenes with different bandgap and a wide adjustable absorption range that can be linearly regulated.

**Band structure analysis**. Based on the experimental results, we established a series of structural models (shown in Supplementary Fig. 8) to investigate the electronic band structure and partial density of states (PDOS) of the proposed 2D gersiloxenes (details in Supplementary Note 3). The theoretical calculations (Supplementary Figs. 9, 10) proved that the gersiloxenes are all direct-bandgap semiconductors. And the calculated bandgap values agree well with the experimentations (Supplementary Fig. 11). The PDOS results suggest that the electronic states of the CB and VB near the Fermi level are hybridized by different orbitals of Ge and Si, O, and H (as summarized in supplementary Table 2 and Supplementary Table 3), which is beneficial for the migration of photogenerated electrons and can suppress the recombination of photogenerated electrons and holes. Moreover, the distribution of the valence band maximum (VBM) and conduction band minimum (CBM) for the gersiloxene with $x = 0.5$ (i.e., HGeSiOH) is similar to that of type-II heterostructures, and the photoinduced electrons and holes would transfer to the CBM (Si) and VBM (Ge), respectively, which is more conducive to the separation of excited electrons and holes.

To determine the band positions of all samples, we investigated the VB positions of all samples by valence-band XPS (VB-XPS). As shown in Supplementary Fig. 12, the VB edge positions of samples with $x = 0.1, 0.3, 0.5, 0.7,$ and $0.9$ are determined to be 0.84, 0.91, 1.14, 1.43 and 1.22 eV, respectively; whereas that of GeH and $Si_6H_3(OH)_3$ are 0.63 and 1.05 eV, respectively. According to the bandgap results obtained from DRS, the position of the conduction band maximum is calculated to be $-0.96$, $-1.04, -1.16, -1.10,$ and $-1.35$ eV for gersiloxenes ($x = 0.1-0.9$), respectively; for GeH and $Si_6H_3(OH)_3$ the calculated values are $-0.90$ and $-1.40$ eV respectively. The resulting electronic band structures are shown in Fig. 5d (The reaction energy levels for the transformation of $CO_2$ into CO and water reduction ($H^+/H_2$) and oxidation ($O_2/H_2O$) are taken from the literature[56]). It is obvious that the CB and VB energies of all gersiloxenes with different $x$ values meet the requirements of photocatalysts for hydrogen production and $CO_2$ reduction to CO. With increasing $x$, the CB

position moves upward (becomes more negative), while the VB position moves downward (becomes more positive), which enables the gersiloxenes to more efficiently drive $H^+$ or $CO_2$ reduction and the oxidation of $H_2O$. For comparison, the energy band structure of GeH is not enough to satisfy the overall reaction conditions for water splitting or $CO_2$ reduction unless a sacrificial agent is used in the photocatalytic reaction process. $Si_6H_3(OH)_3$ can drive photocatalytic oxidation-reduction reactions, but its light absorption range is much smaller than that of GeH and HGeSiOH.

**Photoreduction activity of the resulting materials**. After successful synthesis of bandgap-tunable 2D gersiloxenes and illustration of their electronic structures, the photocatalytic capabilities of gersiloxenes were first evaluated by detecting their hydrogen evolution activity in water with the addition of sodium sulfite (0.10 M) as a sacrificial reagent. The $H_2$ evolution rates (HERs) were measured under 1 bar pressure and irradiation with a 300 W light source. Figure 6a shows the time-dependent photocatalytic hydrogen evolution for all gersiloxenes. As expected, the gersiloxene with $x = 0.5$ exhibited a maximum yield of 9.50 mmol $g^{-1}$ for 6 h, corresponding to an HER of 1.58 mmol $g^{-1}$ $h^{-1}$, which is much higher than that of GeH and $Si_6H_3(OH)_3$. The HERs of gersiloxene increased with $x$ values from 0.17 ($x = 0.1$) and 0.42 ($x = 0.3$) to 1.58 mmol $g^{-1}$ $h^{-1}$ ($x = 0.5$) and then decreased to 0.98 ($x = 0.7$), and 0.78 mmol $g^{-1}$ $h^{-1}$ ($x = 0.9$) (Fig. 6b). Cycling tests showed that the $x = 0.5$ sample had stable activity for the production of hydrogen with little reduction in the hydrogen production rate (Fig. 6c). Furthermore, a photocatalytic $CO_2$ reduction performance test was conducted under an atmosphere with $CO_2/H_2O$. It was found that in a gas-liquid-solid reaction system, CO was the only direct product, and no other carbonaceous products or $H_2$ were detected. Comparison experiments were performed with no light irradiation, with no catalyst, and in the presence of $N_2$ gas. No carbonaceous products were detected (Supplementary Fig. 13), which confirmed that CO evolved from the photocatalytic effect of the gersiloxene in the mixture of $CO_2$ and $H_2O$ (details in Supplementary Note 4). The trend of time-dependent photocatalytic CO evolution (Fig. 6d) is very similar to that of $H_2$ evolution. The gersiloxene with $x = 0.5$ showed a maximal CO evolution rate (COER) of 6.91 mmol $g^{-1}$ $h^{-1}$ (Fig. 6e), in contrast, much higher than that of GeH (1.02 mmol $g^{-1}$ $h^{-1}$) and $Si_6H_3(OH)_3$ (1.51 mmol $g^{-1}$ $h^{-1}$). The COER of gersiloxene with $x = 0.5$ is almost 691 times that of palladium-decorated silicon–hydride nanosheets (Pd@SiNS) tested under 170 °C in $H_2/CO_2$ atmosphere[20], ~28 times that of surface hydride-functionalized silicon nanocrystals (ncSi:H)[57], and higher than overwhelming majority of the photocatalysts reported to date (Supplementary Table 5). And it maintained stable activity for 10 h (Fig. 6f). In addition, the apparent quantum efficiency (AQE) of gersiloxene with $x = 0.5$ measured at different excitation wavelength shows a highest AQE of up to 5.95% at 420 nm (Supplementary Table 4 and Supplementary Fig. 14, details in Supplementary Note 5), which further indicating the high activity of photocatalyst.

The photogenerated carrier recombination and separation behaviors of 2D gersiloxenes were further investigated based on photoluminescence (PL) emission spectroscopy. It can be seen from Fig. 6g that all gersiloxenes have a wide range of fluorescence emissions in the range of 440−650 nm. The larger portion of emission signals at 440−530 nm is ascribed to the free exciton from electron–hole recombination. While the peak at 550−600 nm corresponded to the recombination of two-electron-trapped oxygen vacancies with photogenerated holes. In addition, the PL intensity shows an obvious decrease from $x = 0.1$ to $x = 0.5$ and a subsequent increase from $x = 0.5$ to $x = 0.9$. For

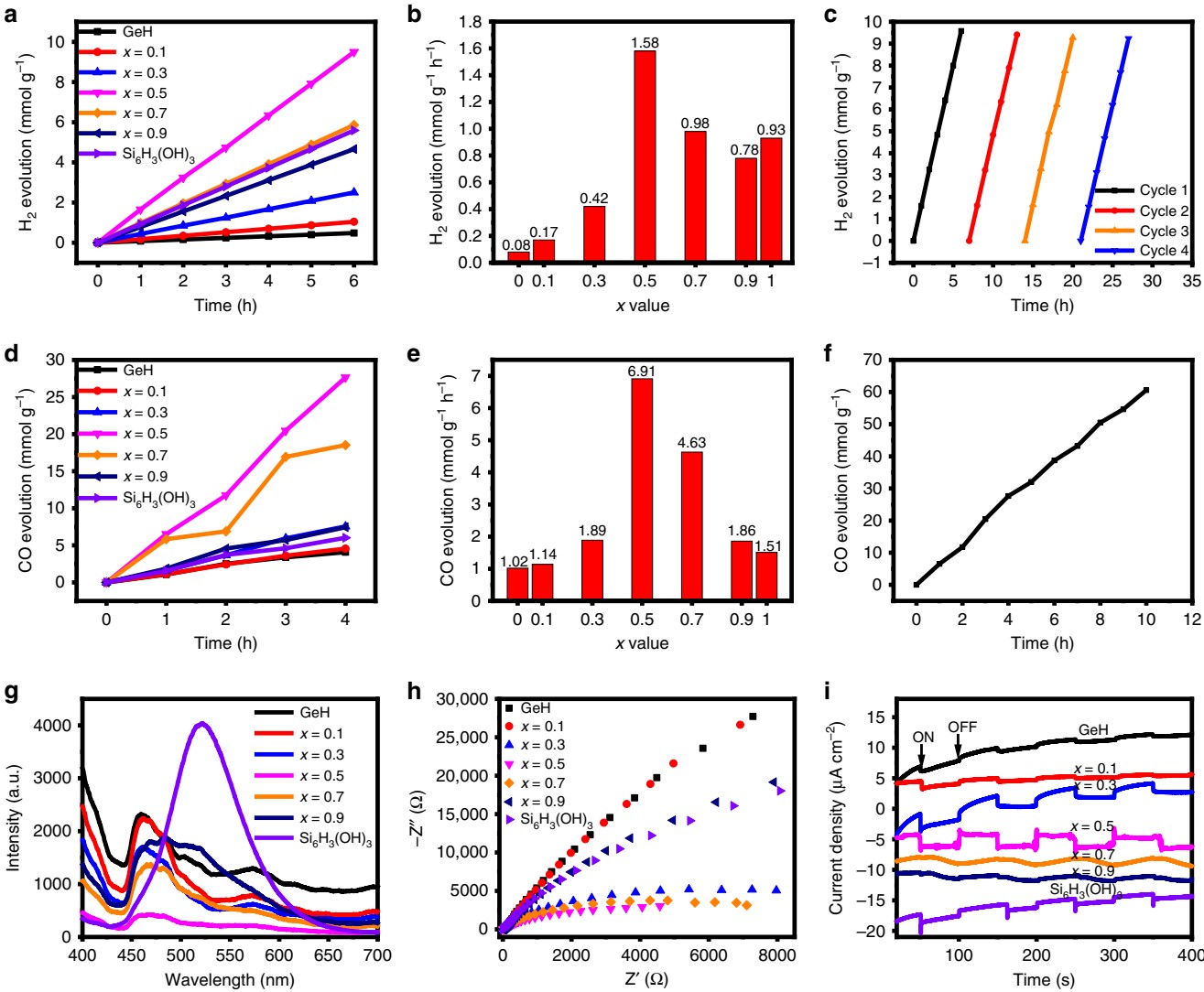

**Fig. 6 Photocatalytic performance and electrochemical characterization. a, b** Time-dependent photocatalytic hydrogen evolution (**a**) and HERs (**b**) of gersiloxenes with $x = 0.1-0.9$, GeH and $Si_6H_3(OH)_3$. **c** Photostability for $H_2$ production of the gersiloxene with $x = 0.5$ (HGeSiOH). **d, e** Time-dependent photocatalytic CO evolution (**d**) and COERs (**e**) of gersiloxenes with $x = 0.1-0.9$, GeH and $Si_6H_3(OH)_3$. **f** Time-dependent photocatalytic CO evolution for 10 h. **g–i** PL spectra (**g**), EIS Nyquist plots (**h**) and transient photocurrent responses (300 W xenon lamp) (**i**) of gersiloxenes with $x = 0.1-0.9$, GeH and $Si_6H_3(OH)_3$. HERs, $H_2$ evolution rates. COERs CO evolution rates, PL photoluminescence, EIS electrochemical impedance spectroscopy. Source data are provided as a Source Data file.

comparison, the PL intensity of GeH and $Si_6H_3(OH)_3$ are much higher than that of the gersiloxene with $x = 0.5$. These results demonstrate that the gersiloxene with $x = 0.5$ has better capability to effectively suppress the carrier recombination than other gersiloxenes, GeH and $Si_6H_3(OH)_3$. Thus, the extensive light absorption ability together with the less radiative electron–hole recombination endowed the gersiloxene ($x = 0.5$, HGeSiOH) with significantly enhanced photoactivity and phooelectrochemical performance. A photoelectrochemical test further confirmed that HGeSiOH shows the best photocurrent performance and the lowest interfacial resistance of charge carriers (Fig. 6h, i), coinciding with the HER and COER results. As HGeSiOH has the highest surface area of 319.7 $m^2 g^{-1}$, the distinctive two-dimensional nanostructure and large specific surface area of HGeSiOH provide more paths for the migration of photogenerated carriers, promote the migration of photogenerated electrons and holes between layers, and inhibit the recombination of carriers, thus greatly improving the photocatalytic performance of materials.

We also performed an electron spin resonance (ESR) test on the sample. ESR spectra (Supplementary Fig. 15a, b) indicate that HGeSiOH has abundant oxygen vacancies and changes little before and after illumination (This further explains the results of oxygen vacancies in XPS, UV, and the PL characterization). These holes may be caused by either Ge or Si atoms because oxygen vacancies are also present in GeH (caused by oxidation) and $Si_6H_3(OH)_3$. Obviously, the concentration of oxygen vacancies in HGeSiOH is higher than that of GeH and $Si_6H_3(OH)_3$. The presence of surface oxygen vacancy defects leads to electron enrichment, which enhances the activation of $CO_2$ molecules, thereby further promoting $CO_2$ reduction[58,59].

**Adsorption energies analysis.** To further investigate the origins of the enhanced photocatalytic performance of the 2D HGeSiOH alloy, we performed DFT calculations of the adsorption energies ($E_{ads}$) of $H_2O$ and $CO_2$ molecules at HGeSiOH, $Si_6H_3(OH)_3$, and GeH surfaces because the adsorption capacity of a catalyst for $H_2O$ or $CO_2$ is one of the most critical factors affecting photocatalytic

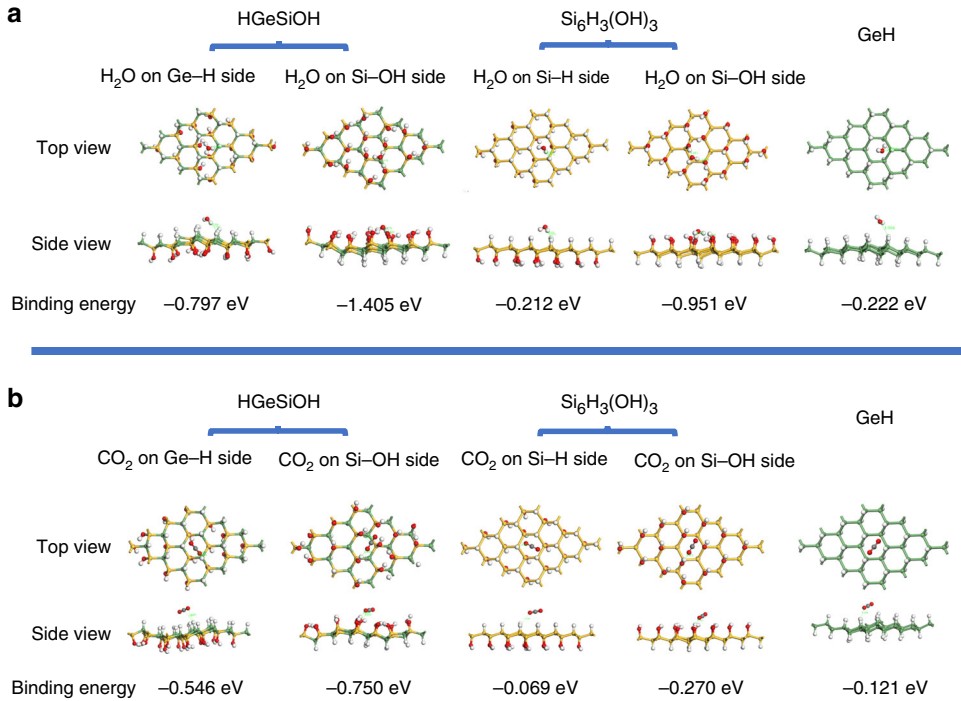

**Fig. 7 The DFT calculations on adsorption energies. a, b** Optimized geometric structures and binding energies for $H_2O$ (**a**) and $CO_2$ (**b**) adsorption on HGeSiOH, $Si_6H_3(OH)_3$, and GeH monolayers.

hydrogen evolution and $CO_2$ reduction performance[60,61]. As Fig. 7a, b shows, the binding of $H_2O$ or $CO_2$ to HGeSiOH and $Si_6H_3(OH)_3$ was considered from two sides of the monolayer structures. For the Si−OH side of HGeSiOH and $Si_6H_3(OH)_3$, the binding energies for $H_2O$ are −1.405 eV and −0.951 eV, respectively. For the other side of HGeSiOH and $Si_6H_3(OH)_3$, the binding energies are −0.797 eV and −0.212 eV, respectively. The binding energy of $H_2O$ to GeH is −0.222 eV. Therefore, the interactions of an $H_2O$ molecule with the HGeSiOH, $Si_6H_3(OH)_3$ and GeH surfaces are energetically favorable, and regardless of which side of the single-layer structure, the $H_2O$ adsorption energy on HGeSiOH is more negative than that on $Si_6H_3(OH)_3$ and GeH, suggesting that the alloyed structure of 2D GeSi can significantly improve its $H_2O$ adsorption capacity. For $CO_2$ adsorption, the binding energies to the Si−OH side of HGeSiOH and $Si_6H_3(OH)_3$ are −0.750 eV and −0.270 eV, respectively. On the other side, the binding energies are −0.546 eV and −0.069 eV, respectively. The binding energy of $CO_2$ to GeH is −0.121 eV, indicating that compared with 2D $Si_6H_3(OH)_3$ and GeH, the alloy-structured 2D HGeSiOH has enhanced $CO_2$ adsorption capacity. In addition, the calculation results show that the atoms closest to $H_2O$ and $CO_2$ on HGeSiOH, $Si_6H_3(OH)_3$, and GeH are H atoms from Ge−H and Si−OH bonds, Si−H and Si−OH bonds, and Ge−H bond, respectively. These findings indicated that both $H_2O$ and $CO_2$ may be adsorbed and activated at the −H and −OH sites and that −OH shows a better ability to interact with $H_2O$ and $CO_2$ (the adsorption energies on the Si−OH side are more negative than those on the Si−H or Ge−H sides). One can clearly observe from the calculated results that Si−OH has a strong interaction with $CO_2$ and $H_2O$, while Si−H and Ge−H form weak coordination interactions with both $CO_2$ and $H_2O$. Therefore, it can be inferred that in the process of photocatalytic $CO_2$ or $H_2O$ reduction, $CO_2$ or $H_2O$ are mainly adsorbed and activated at the −OH site to obtain electrons and further reduced to $H_2$ and CO. A comparison of adsorption energy calculated by different calculation software is presented in Supplementary

Fig. 16 (details in Supplementary Note 6), and the conclusion is the same. The theoretical results align well with our experimental evidences in the fact that HGeSiOH showed better photocatalytic activity than that of GeH and $Si_6H_3(OH)_3$.

In conclusion, we synthesized free-standing two-dimensional honeycomb-like Ge−Si alloy compounds (gersiloxenes) for the first time. The experimental results and theoretical calculations show that these two-dimensional compounds are all direct-bandgap semiconductors with tunable bandgaps ranging from 1.8 to 2.57 eV by increasing the Si content from 10 to 90%. In addition, the appropriate energy band position endows these materials with photocatalytic properties in $H_2O$ reduction to $H_2$ and $CO_2$ reduction to CO. The synthesized gersiloxene HGeSiOH ($x = 0.5$) has a relatively wide spectral response range and a suitable band position, which is most beneficial for the photogeneration of electrons and holes. The sample's relatively higher specific surface area (319.7 $m^2\,g^{-1}$) can provide more pathways for the migration of photogenerated carriers, which improves its photocatalytic performance. DFT calculations prove that the structure of HGeSiOH enhanced the material's $H_2O$ and $CO_2$ adsorption capacity, which is beneficial to the photocatalytic reaction. The hybridized orbital composition and distribution of the VB and CB for HGeSiOH are beneficial for the migration of photogenerated carriers and suppressing the recombination of photogenerated electrons and holes. Moreover, the presence of surface oxygen vacancy defects leads to electron enrichment and enhances the activation of $CO_2$ molecules, facilitating the $CO_2$ reduction. When used as a photocatalyst in water reduction to produce $H_2$, it has a hydrogen production rate of 1.58 mmol $g^{-1}\,h^{-1}$ and good cycle stability. When applied to photoreduction of $CO_2$ under mild conditions (25 °C, 1 atm $CO_2$), it achieves high conversion efficiency to produce CO with a production rate of 6.91 mmol $g^{-1}\,h^{-1}$ and has a high AQE of 5.95% at 420 nm, which is better than the majority of recently reported photocatalysts. We believe that this research will provide many valuable references for the synthesis,

design, regulating electronic properties of new germanium/silicon-based two-dimensional materials and the development of their applications in photocatalysis.

## Methods

**Synthesis of CaGe$_{2-2x}$Si$_{2x}$ crystals.** In a typical reaction, calcium (Ca, 99.5%, Aladdin), germanium (Ge powder, 99.999%, Macklin), and silicon (Si, 99.99%, Adamas) were loaded in stoichiometric amounts into a quartz tube in an argon filled glovebox, then vacuum sealed using a MRVS-1002 vacuum sealing system (Partulab, Wuhan, China), annealed at 1000−1200 °C for 16−20 h with a tube furnace, and cooled to room temperature over 1−5 days.

**Synthesis of Ge$_{1-x}$Si$_x$H$_{1-y}$(OH)$_y$.** CaGe$_{2-2x}$Si$_{2x}$ crystals were stirred in concentrated hydrochloric acid (HCl, AR, 37%) for 3−10 days at −30 °C under argon conditions. After that, the resulting products were transferred to a glove box filled with argon and filtered, followed by washing with Milli-Q H$_2$O and isopropyl alcohol (IPA, C$_3$H$_8$O, AR). Subsequently, the products were redispersed in IPA for 2 h of ultrasonic treatment and then centrifuged for 30 min at a speed of 1000 r min$^{-1}$ to further exfoliate the stacked gersiloxenes. Finally, the products were dried at room temperature on a Schlenk line and kept in an Ar-filled glovebox.

**Characterization.** Flat plane and capillary mode XRD was performed on a D8 Advanced X-ray diffractometer (BRUKER AXS GMBH) under Cu Kα radiation at a wavelength of 0.154 nm. The lattice parameters of the Zintl phases were obtained through Rietveld analysis of the experimental data. FTIR was carried out on an FTIR 650 spectrometer (BRUKER AXS GMBH) with KBr discs. Raman spectra were acquired with a DXR Raman microscope from Thermo Scientific using a 532 nm laser source. Solid-state diffuse reflectance spectra (DRS) were recorded with a Lambda 750 UV/Vis/NIR spectrophotometer (Perkin Elmer) by mixing samples thoroughly with barium sulfate (BaSO$_4$, SP, Rhawn) powder and spreading them evenly on a BaSO$_4$ background holder. The Kubelka-Munk remission function was employed to convert the measured reflectance to absorption. XPS measurements were performed at a power of 450 W using a PHI 1600 surface analysis system equipped with a Mg Kα anode. Field-emission scanning electron microscopy (FESEM) images were recorded by a Hitachi S-4800 field-emission scanning electron microscope equipped with an energy-dispersive X-ray spectrometer (EDS). TEM images and SAED patterns were collected with a JEOL JEM-2100F field-emission Electron Microscope. PL emission spectroscopy were recorded with a Hitachi F-4600 fluorescence spectrophotometer at room temperature. For specific surface area measurements, nitrogen sorption analyses were carried out at 77 K for a P/P$_0$ of 0.025−0.3 with a Micrometrics ASAP 2050 analyzer (Micrometrics Instrument Corporation, Norcross, GA). A Dimension ICON-PT (BRUKER AXS GMBH) AFM was employed to investigate the morphology and thickness of samples. Low-temperature ESR spectra were collected using a BRUKER A300 ESR spectrometer (77 K, 9.063 GHz, X-band). A 300 W Xe lamp was used as a light source.

**Electrochemical analysis.** Electrochemical measurements were performed at room temperature on an electrochemical workstation (CHI 660d) with a three-electrode cell (using a 0.5 M Na$_2$SO$_4$ electrolyte). The working electrodes were prepared by dropping the sample slurry (5 mg dispersed in 200 μL of 0.5 wt% Nafion in ethanol) on a 1.0 cm$^2$ PET ITO substrate and then drying in air at 80 °C. A Pt plate was used as the counter electrode, and an R030 Ag/AgCl electrode (3.5 M KCl) was used as the reference electrode. Electrochemical impedance spectroscopy (EIS) measurements were performed in the frequency range from 100 mHz to 1 MHz with an AC voltage of 5 mV and a bias DC voltage of 0.6 V. Transient photocurrent responses (I–t curves) were tested under 300 W xenon lamp irradiation at 0.05 V.

**Photocatalytic hydrogen evolution test.** Photocatalytic hydrogen evolution experiments were performed in an online photocatalytic hydrogen generation system (CEL-SPH2N-D9, AuLight, Beijing) at ambient temperature (25 °C). In a typical procedure, 50 mg of catalyst was suspended in 100 mL of sodium sulfite solution (Na$_2$SO$_3$, 0.10 M). The suspension was sonicated in an ultrasonic bath for 30 min and then degassed with a vacuum pump for 1 h to completely remove the dissolved oxygen and to ensure that the reaction system was under inert conditions. The produced hydrogen was analyzed by gas chromatography (GC-7920) using a TCD with nitrogen as a carrier gas. A 300 W Xe lamp (CEL-HXF 300, ≈100 mW cm$^{-2}$) was used as the light source.

**Photocatalytic CO$_2$ reduction test.** Photocatalytic CO$_2$ reduction experiments were performed with a Labsolar-6A system (Perfect Light Co., China) at ambient temperature (25 °C). Normally, 60 mg of photocatalyst was ultrasonically dispersed in 50 ml of deionized water containing sodium sulfite (Na$_2$SO$_3$, 0.10 M), and then the suspension was transferred into a reactor. The reactor is connected to a glass-enclosed gas circulation system. First, the reaction vessel was purged with high-purity CO$_2$ for a period, followed by vacuum treatment. Then, the reaction vessel

was purged with high-purity CO$_2$ to reach atmospheric pressure (101 kPa). The whole reaction system is sealed, and the catalyst in the suspension system is continuously stirred with a magnetic stirrer to reach the adsorption-desorption equilibrium of the catalyst in an atmosphere with CO$_2$/H$_2$O to ensure the complete adsorption of gas molecules. At this time, the circulating gas pump is opened to circulate the gas in the reaction vessel, ensuring uniform distribution of the products. A 300 W xenon lamp (24 mW/cm$^2$) was used to illuminate the suspension vertically downward from the top of the reactor. To keep the temperature of the reactor constant, circulating water was used to remove the heat of the reaction. During the reaction, the products in the mixed gas were analyzed every hour by a gas chromatograph (GC-2010 Plus, SHIMADZU, Japan) equipped with a flame ionization detector (FID) and a thermal conductivity detector (TCD). The possible liquid products were detected by liquid chromatography-mass spectrometry (BRUKER ESQUIRE-LC, USA).

**AQE test for photocatalytic CO$_2$ reduction.** The photocatalytic activity of the catalyst at a specific excitation wavelength is obtained by a specific band pass filter (λ = 380, 420, 525, 700 nm) with a 300 W Xe lamp in the same apparatus as the photocatalytic CO$_2$ reduction reaction. Generally, the AQE is calculated as follows:

$$AQE(\%) = \frac{\text{number of the electrons taking part in reaction}}{\text{number of incident photons}} \times 100\% \quad (3)$$

For photocatalytic reduction of CO$_2$ to CO, the AQE is calculated by the following equations,

$$AQE(\%) = \frac{2n(CO)}{I} \times 100\% \quad (4)$$

where $n(CO)$ represents the number of molecules of CO production, and $I$ is the number of incident photons. In actual measurement, it is assumed that all incident photons are absorbed by the suspension; the calculation formula of the incident photon number $I$ is as follows:

$$I = PSt\frac{\lambda}{h\nu} \quad (5)$$

where $P$ is the average intensity of the radiation (W/cm$^2$), $S$ is the incident irradiation area ($S = 19.63$ cm$^2$), $t$ is the irradiation time ($t = 4 \times 3600$ s), $\lambda$ is the incident wavelength, $h$ is the Planck constant ($6.626 \times 10^{-34}$ J s), and $\nu$ is the speed of light ($3 \times 10^8$ m s$^{-1}$). Therefore, the calculation equation of AQE for photocatalytic reduction of CO$_2$ to CO can be written as:

$$AQE(\%) = \frac{2CN_A}{PSt \times \frac{\lambda}{h\nu}} \times 100\% \quad (6)$$

where $C$ is the CO production amount (mol); $N_A$ is the Avogadro constant ($6.02 \times 10^{23}$ mol$^{-1}$).

**Computational analysis for band structure.** Density functional theory (DFT) calculations were performed with the Vienna Ab initio Simulation Package (VASP) using the projector augmented wave (PAW) Perdew-Burke-Ernzerhof (PBE) pseudopotentials of the generalized gradient approximation (GGA). For monolayer structures, the calculations were performed in a fully relaxed manner until the convergence accuracy of the force on each atom was <10$^{-3}$ eV/Å and the energy convergence accuracy was <10$^{-5}$ eV. The vacuum height was set as >15 Å to eliminate the interactions between neighboring monolayers. A Monkhorst-Pack grid of (11 × 11 × 1) k points was used for electronic property calculations and geometry optimizations. The Heyd-Scuseria-Ernzerhof hybrid functional (HSE06) was used to calculate the band structures. The planewave cutoff energy was set to 400 eV. For the two-layer unit cell of bulk structures, the optimizations were carried out until the maximum force upon each relaxed atom was <−0.01 eV/Å and the energy convergence accuracy was <10$^{-4}$ eV. A Monkhorst-Pack grid of (5 × 1 × 3) k points was used for electronic property calculations and geometry optimizations. HSE06 was used to calculate the band structures, and the planewave cutoff energy was set to 500 eV.

**Calculations of adsorption energy.** The interactions of HGeSiOH, Si$_6$H$_3$(OH)$_3$ and GeH with H$_2$O and CO$_2$ were calculated by DFT. The Dmol3 module in Materials Studio software was used for calculation. HGeSiOH, Si$_6$H$_3$(OH)$_3$ and GeH monolayer structures were all modeled as 4 × 4 superlattices, and the interlayer spacing between different layers was set to 20 Å to avoid the influence of interaction forces between different layers. In the process of structural optimization, the k-point of the Brillouin zone was set to 7 × 7 × 1. The GGA with the PBE functional was used as the exchange correlation function, the double numerical plus (DNP) polarization function was applied for the basis set of all the electrons, and the all-electron relativistic method was used to deal with the internal nuclear electrons. The convergence accuracy of the energy, maximum force and maximum displacement was set to $1 \times 10^{-5}$ Ha, $2 \times 10^{-3}$ Ha/Å (1 Ha = 27.2114 eV), and $5.0 \times 10^{-3}$ Å, respectively. Moreover, for comparison, we used VASP to perform the calculations of adsorption energy using the same parameter as the Dmol3 module in Materials Studio software, this would be helpful for the interest of relevant researchers. The simulated adsorption energies (E$_{ads}$) of H$_2$O and CO$_2$

molecules on $HGeSiOH$, $Si_6H_3(OH)_3$ and $GeH$ surfaces were calculated by the following equation:

$$E_{ads} = E_{total} - (E_m + E_n), \left(m : HGeSiOH/Si_6H_3(OH)_3/GeH; n : H_2O/CO_2\right) \tag{7}$$

where $E_{total}$ represents the total energy of $H_2O$ or $CO_2$ adsorbed onto $HGeSiOH$, $Si_6H_3(OH)_3$, or $GeH$ monolayers, $E_m$ stands for the energy of pristine $HGeSiOH$, $Si_6H_3(OH)_3$, or $GeH$ monolayers, and $E_n$ stands for the energy of $H_2O$ or $CO_2$ molecules.

## Data availability

The data that support the findings of this study are available from the corresponding author upon reasonable request.

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

## Acknowledgements

This work was financially supported by National Key R&D Program of China (No. 2016YFA0202302), the State Key Program of National Natural Science Foundation of China (No. 51633007), National Natural Science Funds for Distinguished Young Scholars (No. 51425306), and National Natural Science Foundation of China (No. 51573125 and 51773147).

## Author contributions

F. Zhao, Y.F., and W.F. conceived the experiment. F. Zhao and Y.W. synthesized the $CaGe_{2-2x}Si_{2x}$ crystals and 2D gersiloxenes. F. Zhao, X.Z., X.L., characterized the physical and chemical properties of as-prepared samples involving XRD, DRS, FTIR, Raman, FESEM, TEM, AFM, XPS, BET, PL and ESR, assisted by F. Zhang. and Z.L. F. Zhao and X.Z. conducted the electrochemical measurements. F. Zhao and Y.W. performed the theoretical calculations. T.W. and J.G. performed and analysed the photocatalytic results and provided theoretical support. F. Zhao wrote the manuscript with contributions from all authors. All the authors discussed the results.

## Competing interests

The authors declare no competing interests.
