## [Peer Review File · Nature Communications]

Reviewers' comments:

Reviewer #1 (Remarks to the Author):

1) General remarks:

The authors have synthesized two-dimensional silicone-germanium alloys containing terminal H- and OH-bonds by a convenient solid-state reaction. Impressive physical characterization and theoretical calculations support the structural and electronic properties.

Problematic is the discussion of the photocatalysis part of the manuscript. Since water or carbon dioxide can be photoreduced (that is no splitting of water, what means formation of hydrogen and oxygen) at almost any semiconductor. The novel materials just exhibit expected properties. Furthermore, discussion of UV-Vis spectra and differences in photocatalytic activities is oversimplified and neglects detailed present mechanistic knowledge in semiconductor photocatalysis.

Considering all these aspects, I cannot recommend acceptance. Recommendable is submission of a full paper to a journal focusing on solid state chemistry.

2) Special remarks:

Abstract and introduction are almost identical.

a) The maximal product formation rates calculated per gram of photocatalyst do not allow a reliable comparison of photocatalytic activities (see recent textbook on Semiconductor Photocatalysis by H. Kisch).

b) Hydrogen formation in presence of a reducing agent (sodium sulfite) is NOT water splitting, but simple water reduction.

Lines 244-253:

These are too speculative assumptions. At least at this place of the text.

Line 112-141:

a) How many milligrams of starting materials were used? What was the weight of product obtained?

b) How was product purity proven?

c) Are no amorphous by-products formed?

d) What is the experimental error in the lattice constant measurements?

Lines 243-244:

Giving two decimal places for the specific surface seems a bit overdone

Lines 247-253:

Here are unnecessary, too speculative assumptions made. At least at this place of the text.

Lines 257-270:

a) Discussion of the DRS is too superficial: almost all samples (especially the one with $x=0.5$) exhibit pronounced sub-bandgap absorptions which may be of crucial influence on the photocatalytic activity.

b) Bandgaps obtained from the depicted Tauc plots should have a rather large experimental error. What was the reproducibility?

c) Comparison of weight-related reaction rates is not reliable (see first remark above)

Reviewer #2 (Remarks to the Author):

This work reports a new 2D GeSi alloy with enhanced photocatalytic performance for H₂ evolution and CO₂ photoreduction to CO. By means of topotactic deintercalation process the authors synthesized a two-dimensional honeycomb-like Ge-Si alloy compounds terminated by -H/-OH (gersiloxenes). The experimental results and theoretical calculations confirmed the gersiloxenes are all direct-bandgap semiconductors with monotonical increase of bandgap (1.8-2.57 eV) as Si content x increase ($x = 0.1-0.9$). Among the synthesized gersiloxene the HGeSiOH ($x=0.5$) has highest specific surface area

(319.7 m² g⁻¹) and shows excellent photocatalytic capabilities, providing a new possible option in the field of catalysts for H₂ evolution and CO₂ photoreduction to CO.

The paper presents the detailed characterizations of synthesized samples to measure various physical and chemical properties, electrochemical and photocatalytic data. The theoretical calculations of DFT for band structures and adsorption energy of H₂O and CO₂ on HGeSiOH, Si₆H₃(OH)₃ and GeH have been conducted to understand in depth the photocatalytic behaviors of samples. Some discussions are as followed.

1. In the manuscript "The theoretical calculations (Supplementary Fig. 8 and 9) proved that the gersiloxenes are all direct bandgap semiconductors. And the calculated bandgap values agree well with the experimentations." Pick the experimental bandgap data in the manuscript and calculational bandgap data in the supplementary, draw them in the chart as following. (please find the chart in the attachment - comment for author) The trends are similar except the case at x=0.3. Please check or calculate again the data at x=0.3 carefully. Another question: what is the authors' purpose to calculate the two layers (bulk) model, which is not discussed in the manuscript/ supplementary.

2. Just above the conclusion in the manuscript "These holes may be caused by either Ge or Si atoms because oxygen vacancies are also present in GeH and Si₆H₃(OH)₃ (Supplementary Fig. 12)." How to understand that the oxygen vacancies present in GeH? Please provide clear description.

3. In this work the calculations of band structure are based on VASP, while the calculations of adsorption energy are base on Dmol3 module in Materials Studio software. How about directly use VASP to perform the calculations of adsorption energy (which material researchers often do)? What is the authors' consideration? Comparison or discussion would be helpful for the interest of relevant researchers.

Reviewer #3 (Remarks to the Author):

This report by Zhao et al. entitled "Novel two-dimensional gersiloxenes with tunable bandgap for photocatalytic H₂ evolution and CO₂ photoreduction to CO" synthesizes 2D -H/-OH terminal substituted siligenes, which they term as gersiloxnes. These are made from CaGe_{2-2x}Si_{2x}. The 2D materials are characterized by XRD, TEM, AFM, FTIR, Raman, XPS, UV-Vis diffuse reflectance spectroscopy. The authors show photocatalytic activity for H₂ and CO generation. Furthermore, the authors rely on DFT calculations to determine the electronic band structure. This is an interesting material system that seems to be promising and would be of wide interest. I recommend that this paper be reconsidered after major edits.

First, the figures are very small and I am not confident in the author's analysis because the figure size is too small. Also, the color choices and line widths need to be selected such that they can be easily interpreted. XPS data should be plotted from high to low energy, which is the convention for XPS.

In general, I think that the authors did all of the correct characterization methods and catalysis tests; however, I am not confident in the results. For the XPS data provided in the SI, there appears to be intensity at 0eV for all of the spectra shown in the VB region. Also, the figures should be plotted such that this VB area (-1 to 5 eV) is shown. This would give reviewers confidence (or not) in the fits, which directly provides the data for Figure 5c.

Another example is the photocatalytic testing. The controls should be more clearly discussed with the results shown in SI. No light, no catalyst and in the presence of N₂ gas for the CO₂ photoreduction.

Control samples of x=0 and x=1 should also be shown for the various characterization methods. This would give confidence in the peak assignments.

Figure 4c shows that the samples have Cl. Cl is a great passivation layer. How do the authors know that the surfaces don't have -Cl and only have -O/-OH? I would imagine Cl is present and important.

Also, with Figure 5c, how did the authors determine the energetics for the H⁺ and CO₂ reduction. This should be discussed in the text. Moreover, XPS is performed under ultra-high vacuum and H⁺ and CO₂ reduction is in electrolyte. How are you able to be put on the same energy scale. This should be discussed.

For the DFT calculations, I am surprised that the monolayer band gap isn't consistently larger than the bulk structures. Can the authors comment.

The authors claim that the material has an internal Type II heterostructure between the Ge and Si, which leads to charge separation. I find that this argument is not well supported since the authors argue with SEM elemental mapping that the 2D layers have an even Si/Ge distribution (no grain boundaries) Therefore, how do you get a type II heterostructure. This statement should be discussed further.

Reviewers' comments:

Reviewer #1 (Remarks to the Author):

1) General remarks:

The authors have synthesized two-dimensional silicone-germanium alloys containing terminal H- and OH-bonds by a convenient solid-state reaction. Impressive physical characterization and theoretical calculations support the structural and electronic properties.

Problematic is the discussion of the photocatalysis part of the manuscript. Since water or carbon dioxide can be photoreduced (that is no splitting of water, what means formation of hydrogen and oxygen) at almost any semiconductor. The novel materials just exhibit expected properties. Furthermore, discussion of UV-Vis spectra and differences in photocatalytic activities is oversimplified and neglects detailed present mechanistic knowledge in semiconductor photocatalysis.

Considering all these aspects, I cannot recommend acceptance. Recommendable is submission of a full paper to a journal focusing on solid state chemistry.

2) Special remarks:

Abstract and introduction are almost identical.

a) The maximal product formation rates calculated per gram of photocatalyst do not allow a reliable comparison of photocatalytic activities (see recent textbook on Semiconductor Photocatalysis by H. Kisch).

b) Hydrogen formation in presence of a reducing agent (sodium sulfite) is NOT water splitting, but simple water reduction.

Lines 244-253:

These are too speculative assumptions. At least at this place of the text.

Line 112-141:

a) How many milligrams of starting materials were used? What was the weight of product obtained?

b) How was product purity proven?

c) Are no amorphous by-products formed?

d) What is the experimental error in the lattice constant measurements?

Lines 243-244:

Giving two decimal places for the specific surface seems a bit overdone

Lines 247-253:

Here are unnecessary, too speculative assumptions made. At least at this place of the text.

Lines 257-270:

- a) Discussion of the DRS is too superficial: almost all samples (especially the one with $x=0.5$) exhibit pronounced sub-bandgap absorptions which may be of crucial influence on the photocatalytic activity.
- b) Bandgaps obtained from the depicted Tauc plots should have a rather large experimental error. What was the reproducibility?
- c) Comparison of weight-related reaction rates is not reliable (see first remark above)

Reviewer #2 (Remarks to the Author):

This work reports a new 2D GeSi alloy with enhanced photocatalytic performance for H₂ evolution and CO₂ photoreduction to CO. By means of topotactic deintercalation process the authors synthesized a two-dimensional honeycomb-like Ge-Si alloy compounds terminated by -H/-OH (gersiloxenes). The experimental results and theoretical calculations confirmed the gersiloxenes are all direct-bandgap semiconductors with monotonical increase of bandgap (1.8-2.57 eV) as Si content x increase ($x = 0.1-0.9$). Among the synthesized gersiloxene the HGeSiOH ($x=0.5$) has highest specific surface area (319.7 m² g⁻¹) and shows excellent photocatalytic capabilities, providing a new possible option in the field of catalysts for H₂ evolution and CO₂ photoreduction to CO.

The paper presents the detailed characterizations of synthesized samples to measure various physical and chemical properties, electrochemical and photocatalytic data. The theoretical calculations of DFT for band structures and adsorption energy of H₂O and CO₂ on HGeSiOH, Si₆H₃(OH)₃ and GeH have been conducted to understand in depth the photocatalytic behaviors of samples. Some discussions are as followed.

1. In the manuscript “The theoretical calculations (Supplementary Fig. 8 and 9) proved that the gersiloxenes are all direct bandgap semiconductors. And the calculated bandgap values agree well with the experimentations.” Pick the experimental bandgap data in the manuscript and calculational bandgap data in the supplementary, draw them in the chart as following. (please find the chart in the attachment - comment for author) The trends are similar except the case at $x=0.3$. Please check or calculate again the data at $x=0.3$ carefully. Another question: what is the authors’ purpose to calculate the two layers (bulk) model, which is not discussed in the manuscript/ supplementary.

2. Just above the conclusion in the manuscript “These holes may be caused by either Ge or Si atoms because oxygen vacancies are also present in GeH and Si₆H₃(OH)₃ (Supplementary Fig. 12).” How to understand that the oxygen vacancies present in GeH? Please provide clear description.

3. In this work the calculations of band structure are based on VASP, while the calculations of adsorption energy are based on Dmol3 module in Materials Studio software. How about directly use VASP to perform the calculations of adsorption energy (which material researchers often do)? What is the authors’ consideration? Comparison or discussion would be helpful for the interest of relevant researchers.

Reviewer #3 (Remarks to the Author):

This report by Zhao et al. entitled “Novel two-dimensional gersiloxenes with tunable bandgap for photocatalytic H₂ evolution and CO₂ photoreduction to CO” synthesizes 2D -H/-OH terminal substituted siligenes, which they term as gersiloxnes. These are made from CaGe_{2-2x}Si_{1-2x}. The 2D materials are characterized by XRD, TEM, AFM, FTIR, Raman, XPS, UV-Vis diffuse reflectance spectroscopy. The authors show photocatalytic activity for H₂ and CO generation. Furthermore, the authors rely on DFT calculations to determine the electronic band structure. This is an interesting material system that seems to be promising and would be of wide interest. I recommend that this paper be reconsidered after major edits.

First, the figures are very small and I am not confident in the author’s analysis because the figure size is too small. Also, the color choices and line widths need to be selected such that they can be easily interpreted. XPS data should be plotted from high to low energy, which is the convention for XPS.

In general, I think that the authors did all of the correct characterization methods and catalysis tests; however, I am not confident in the results. For the XPS data provided in the SI, there appears to be intensity at 0eV for all of the spectra shown in the VB region. Also, the figures should be plotted such that this VB area (-1 to 5 eV) is shown. This would give reviewers confidence (or not) in the fits, which directly provides the data for Figure 5c.

Another example is the photocatalytic testing. The controls should be more clearly discussed with the results shown in SI. No light, no catalyst and in the presence of N₂ gas for the CO₂ photoreduction.

Control samples of x=0 and x=1 should also be shown for the various characterization methods. This would give confidence in the peak assignments.

Figure 4c shows that the samples have Cl. Cl is a great passivation layer. How do the authors know that the surfaces don’t have -Cl and only have -O/-OH? I would imagine Cl is present and important.

Also, with Figure 5c, how did the authors determine the energetics for the H⁺ and CO₂ reduction. This should be discussed in the text. Moreover, XPS is performed under ultra-high vacuum and H⁺ and CO₂ reduction is in electrolyte. How are you able to be put on the same energy scale. This should be discussed.

For the DFT calculations, I am surprised that the monolayer band gap isn’t consistently larger than the bulk structures. Can the authors comment.

The authors claim that the material has an internal Type II heterostructure between the Ge and Si, which leads to charge separation. I find that this argument is not well supported since the authors argue with SEM elemental mapping that the 2D layers have an even Si/Ge distribution (no grain boundaries) Therefore, how do you get a type II heterostructure. This statement should be discussed further.

Replies to Reviewers' Comments

We deeply appreciate the reviewer for their careful reading and comments about our manuscript. All their comments have been seriously considered and the updated manuscript gives a clearer and better expression of our work.

Our major revisions include:

1. Measured the apparent quantum efficiency (AQE) for photocatalytic CO₂ reduction to further bolster the high photocatalytic activity. (Reviewer 1)
2. Made deep discussion on the UV-Vis spectra and differences in photocatalytic activities, with XPS, BET, PL, and ESR characterization and photoelectrochemical test to track the photocatalytic activities. (Reviewer 1)
3. Modified the abstract and introduction sections to show the difference. (Reviewer 1)
4. Corrected the wrong description on the photoreduction of water in the manuscript. (Reviewer 1)
5. Recalculated the data of bulk structure at $x=0.3$, and draw the calculational and experimental bandgap data in the chart. (Reviewer 2)
6. Performed the calculations of adsorption energy using VASP. (Reviewer 2)
7. Adjusted the figures to present the convincing results such that they can be easily interpreted. (Reviewer 3)
8. Clearly discussed the results of CO₂ photoreduction under comparative test conditions of “No light, no catalyst and in the presence of N₂ gas”. (Reviewer 3)
9. Added various characterizations for $x=0$ and $x=1$. (Reviewer 3)

Below we discuss each comment in details

Reviewer #1 (Remarks to the Author):

1) General remarks:

The authors have synthesized two-dimensional silicone-germanium alloys containing terminal H- and OH-bonds by a convenient solid-state reaction.

Impressing physical characterization and theoretical calculations support the structural and electronic properties.

Problematic is the discussion of the photocatalysis part of the manuscript. Since water or carbon dioxide can be photoreduced (that is no splitting of water, what means formation of hydrogen and oxygen) at almost any semiconductor. The novel materials just exhibit expected properties. Furthermore, discussion of UV-Vis spectra and differences in photocatalytic activities is oversimplified and neglects detailed present mechanistic knowledge in semiconductor photocatalysis.

Considering all these aspects, I cannot recommend acceptance. Recommendable is submission of a full paper to a journal focusing on solid state chemistry.

Ans 1): Thank you for the insightful comments. We really appreciate your advice and have made a revision accordingly. Firstly, we have corrected the wrong description on the photoreduction of water in the manuscript.

Line 30-30: “All gersiloxenes are direct-gap semiconductors and have wide range of light absorption and suitable band positions for light driven water reduction into H₂ and CO₂ reduction to CO under mild conditions.”

Line 103-104: “Tests for water reduction to H₂ showed that the hydrogen production rate is 1.58 mmol g⁻¹ h⁻¹ and that the material has good cycle stability.”

Line 405-407: “Therefore, it can be inferred that in the process of photocatalytic CO₂ or H₂O reduction, CO₂ or H₂O are mainly adsorbed and activated at the -OH site to obtain electrons and further reduced to H₂ and CO.”

Line 423-425: “In addition, the appropriate energy band position endows these materials with photocatalytic properties in H₂O reduction to H₂ and CO₂ reduction to CO.”

Line 433-442: “When used as a photocatalyst in water reduction to produce H₂, it has a hydrogen production rate of 1.58 mmol g⁻¹ h⁻¹ and good cycle stability. When applied to photoreduction of CO₂ under mild conditions (25 °C, 1 atm CO₂), it achieves high conversion efficiency to produce CO with a production rate of 6.91

mmol g⁻¹ h⁻¹ and has a high AQE of 5.95% at 420 nm, which is better than the majority of recently reported photocatalysts. We believe that this research will provide many valuable references for the synthesis, design, regulating electronic properties of new germanium/silicon-based two-dimensional materials and the development of their applications in photocatalysis.”

Secondly, we are focused on the discussion of UV-Vis spectra and differences in photocatalytic activities, and make deep discussion on the UV-Vis spectra with XPS characterization. In addition, the BET, PL, ESR and photoelectrochemical test are used to track the photocatalytic activities. DFT calculations are also performed to explain the enhanced photocatalytic performance.

Line 212-231: “The high-resolution Ge3d XPS spectra exhibit peaks at ca. 30 eV, corresponding to the Ge-Ge bonds (Fig. 4f), and the shoulder peaks around 32.7eV are ascribed to Ge-O bonds, which is due to the trace oxidation on the surface similar to that of germanium nanosheets.⁵⁵⁻⁵⁶ In the Si2p spectra, the peaks were located at binding energies of approximately 99.9 and 103.0 eV, corresponding to Si-Si bonds and Si-O bonds in the two-dimensional Si chain network (Fig. 4e), respectively. By contrast, the Ge-Ge and Si-Si bonds for the precursor CaGe_{2-2x}Si_{2x} are located at 28.8 and 98.5 eV (Supplementary Fig. 4), respectively. This result indicates that CaGe_{2-2x}Si_{2x} transforms to 2D Ge_{1-x}Si_xH_{1-y}(OH)_y. In addition, the Si2p spectra show that the content of the Si-Si bond relative to that of the Si-O bond in Ge_{1-x}Si_xH_{1-y}(OH)_y increases with an increasing x (when x=0.9, the content of the Si-Si bond is obviously higher than that of the Si-O bond), further indicating that when x > 0.5, each Si atom is terminated not only by OH but also by H. Moreover, O1s spectrum in the range of 527 eV~536 eV exhibits the states of surface oxygen on samples (Fig. 4d). The O1s peaks can be also fitted into several components. Peaks at around 531, 532.4, and 533.1 eV are attributed to the Ge-O, Si-O and O-H, respectively. For GeH, Si₆H₃(OH)₃ and all gersiloxenes, a same fitted peak located at around 531.7 eV, is supposed to be associated with the surface oxygen vacancies,⁵⁷⁻⁵⁸ which would significantly influence their optical properties.”

Fig. 4. (a) FTIR and (b) Raman spectra of 2D $\text{Ge}_{1-x}\text{Si}_x$ alloys with different x values. (c-f) XPS spectra of 2D $\text{Ge}_{1-x}\text{Si}_x\text{H}_{1-y}(\text{OH})_y$ ($x=0.1, 0.3, 0.5, 0.7, 0.9$), GeH and $\text{Si}_6\text{H}_3(\text{OH})_3$. (c) XPS survey spectra. High-resolution XPS spectra of (d) O1s (e) Si2p, and (f) Ge3d.

Line 266-287: “As shown in Fig. 5a, all gersiloxene samples have a wide range of

light absorption from ultraviolet to visible light. The absorption range of gersiloxenes is between that of GeH and $\text{Si}_6\text{H}_3(\text{OH})_3$ (supplementary information Fig. 13a) With the increasing x , the absorption edge shifts to the direction of short wavelength, which is consistent with the change in sample's color from dark red, brick red, brown, yellow green, to light green (Fig. 5a inset). It is worth noting that gersiloxene shows obvious sub-bandgap absorptions compared with pure GeH and pure $\text{Si}_6\text{H}_3(\text{OH})_3$. This sub-bandgap absorption is supposed to be caused by the oxygen vacancies, which is indicated by the XPS O1s spectra. Gersiloxene with $x = 0.5$ exhibits the most significant sub-bandgap absorption, indicating that it has the highest oxygen vacancies concentration than other ones which may be attributed to its graphene-like 2D structure and the highest specific surface area among the as-synthesized gersiloxenes. The significant wide sub-bandgap absorption further increases the light absorption range of the material, which is advantageous for the enhanced photocatalytic and photoelectrochemical performance. Fig. 5b presents the corresponding Tauc plots, which are calculated based on the assumption that the gersiloxenes are direct-bandgap semiconductor materials and proved by subsequent theoretical calculations. As shown in Fig. 5b, with an increasing x , the bandgap increased from 1.8 eV ($x=0.1$) to 2.57 eV ($x=0.9$), indicating that different x values (the content of Si) endowed the 2D gersiloxenes with different bandgap and a wide adjustable absorption range that can be linearly regulated.”

Line 355-377: “The photogenerated carrier recombination and separation behaviors of 2D gersiloxenes were further investigated based on PL spectroscopy. It can be seen from Fig. 6g that all gersiloxenes have a wide range of fluorescence emissions in the range of 440-650 nm. The larger portion of emission signals at 440-530 nm are ascribed to the free exciton from electron–hole recombination. While the peak at 550-600 nm corresponded to the recombination of two \square electron \square trapped oxygen vacancies with photogenerated holes. In addition, the PL intensity shows an obvious decrease from $x=0.1$ to $x=0.5$ and a subsequent increase from $x=0.5$ to $x=0.9$. For comparison, the PL intensity of GeH and $\text{Si}_6\text{H}_3(\text{OH})_3$ (Supplementary Fig. 14a) were also tested, both of which are higher than that of the gersiloxene with $x=0.5$. These

results demonstrate that the gersiloxene with $x=0.5$ has better capability to effectively suppress the carrier recombination than other gersiloxenes, GeH and $\text{Si}_6\text{H}_3(\text{OH})_3$. Thus, the extensive light absorption ability together with the less radiative electron-hole recombination endowed the gersiloxene ($x=0.5$, HGeSiOH) with significantly enhanced photoactivity and photoelectrochemical performance. A photoelectrochemical test further confirmed that HGeSiOH shows the best photocurrent performance and the lowest interfacial resistance of charge carriers (Fig. 6h and I and Supplementary Fig. 14b and c), coinciding with the HER and COER results. As HGeSiOH has the highest surface area of $319.7 \text{ m}^2 \text{ g}^{-1}$, the distinctive two-dimensional nanostructure and large specific surface area of HGeSiOH provide more paths for the migration of photogenerated carriers, promote the migration of photogenerated electrons and holes between layers, and inhibit the recombination of carriers, thus greatly improving the photocatalytic performance of materials.”

Fig. 6. Time-dependent photocatalytic hydrogen evolution (a) and HERs (b) of gersiloxenes with $x=0.1-0.9$. (c) Photostability for H₂ production of the gersiloxene with $x=0.5$ (HGeSiOH). Time-dependent photocatalytic CO evolution (d) and COERs (e) of gersiloxenes with $x=0.1-0.9$. (f) Time-dependent photocatalytic CO evolution for 10 h. PL spectra (g), EIS Nyquist plots (h) and transient photocurrent responses (300 W xenon lamp) (i) of gersiloxenes with $x=0.1-0.9$.

Supplementary Fig. 14 PL spectra (a), EIS Nyquist plots (b) and transient photocurrent responses (300 W xenon lamp) (c) of GeH, $\text{Si}_6\text{H}_3(\text{OH})_3$ and HGeSiOH . (d) HER and COER of GeH and $\text{Si}_6\text{H}_3(\text{OH})_3$.

Supplementary Fig. 17. ESR spectra of GeH, HGeSiOH , and $\text{Si}_6\text{H}_3(\text{OH})_3$ under dark (a) and light (b).

Line 408-416: “We also performed an electron spin resonance (ESR) test on the

sample. ESR spectra indicate that HGeSiOH has abundant oxygen vacancies and changes little before and after illumination (This further explains the results of oxygen vacancies in XPS, UV and the PL characterization). These holes may be caused by either Ge or Si atoms because oxygen vacancies are also present in GeH (caused by oxidation) and $\text{Si}_6\text{H}_3(\text{OH})_3$ (Supplementary Fig. 17). Obviously, the concentration of oxygen vacancies in HGeSiOH is higher than that of GeH and $\text{Si}_6\text{H}_3(\text{OH})_3$. The presence of surface oxygen vacancy defects leads to electron enrichment, which enhances the activation of CO_2 molecules, thereby further promoting CO_2 reduction⁶⁵⁻⁶⁷,

2) Special remarks:

Abstract and introduction are almost identical.

Ans 2): Thank you for your constructive suggestion. We have revised the abstract and introduction.

Line 24-37: “The discovery of graphene and graphene-like two-dimensional materials has brought fresh vitality to the field of photocatalysis. Bandgap engineering has always been an effective way to make semiconductors snugly suitable for specific applications such as photocatalysis and optoelectronics. Achieving control over the bandgap helps to improve the light absorption capacity of the semiconductor materials, thereby improving the photocatalytic performance. This work reported novel two-dimensional -H/-OH terminal-substituted siligenes (gersiloxenes) with tunable bandgap. All gersiloxenes are direct-gap semiconductors and have wide range of light absorption and suitable band positions for light driven water reduction into H_2 and CO_2 reduction to CO under mild conditions. The gersiloxene with the best performance can provide a maximum CO production of $6.91 \text{ mmol g}^{-1} \text{ h}^{-1}$, and a high apparent quantum efficiency (AQE) of 5.95% at 420 nm. This work may open up new insights into the discovery, research and application of new two-dimensional materials in photocatalysis.”

Line 86-106: “In the study, we reported for the first time freestanding siligenes terminated with -H/-OH ($\text{Ge}_{1-x}\text{Si}_x\text{H}_{1-y}(\text{OH})_y$, $x=0.1-0.9$) and named them gersiloxenes,

which were synthesized by the topochemical transformation of freestanding $\text{Ca}(\text{Ge}_{1-x}\text{Si}_x)_2$ alloys prepared by annealing stoichiometric ratios of calcium, germanium and silicon. By combining the experimental results with theoretical calculations, we demonstrated their direct gap type and the bandgap dependence on x (the content of Si), which increase with x values from 1.8 to 2.57 eV. All gersiloxenes have appropriate energy band positions that can drive reduction and oxidation reactions for photocatalysis. Among the gersiloxenes synthesized, the gersiloxene with $x=0.5$ (HGeSiOH) has the theoretical VBM and CBM distribution similar to that of type-II heterostructures, which is more conducive to the separation of excited electrons and holes, and it has a wide spectral response range from the ultraviolet to the near infrared region due to the existence of oxygen vacancies, which is most beneficial for the photogeneration of electrons and holes. In addition, the high specific surface area ($319.7 \text{ m}^2 \text{ g}^{-1}$) may provide more paths for the migration of photogenerated carriers, which can greatly improve photocatalytic performance. Moreover, the presence of surface oxygen vacancy defects leads to electron enrichment and enhances the activation of H_2O or CO_2 molecules, facilitating the release of protons or the reduction of CO_2 . Tests for water reduction to H_2 showed that the hydrogen production rate is $1.58 \text{ mmol g}^{-1} \text{ h}^{-1}$ and that the material has good cycle stability. Impressively, when gersiloxene is used as a photocatalyst to reduce CO_2 under mild conditions ($25 \text{ }^\circ\text{C}$, 1 atm CO_2), the CO production rate is as high as $6.91 \text{ mmol g}^{-1} \text{ h}^{-1}$.”

a) The maximal product formation rates calculated per gram of photocatalyst do not allow a reliable comparison of photocatalytic activities (see recent textbook on Semiconductor Photocatalysis by H. Kisch).

Ans 3): Thank you for your constructive suggestion. We have carefully read the recent textbook on Semiconductor Photocatalysis by H. Kisch (Semiconductor Photocatalysis - principles and applications, John Wiley & Sons, 2015).

The book gives the mechanisms, kinetics, rates, quantum yields, and their comparability of semiconductor photocatalysis. It concluded from the foregoing

discussion that the comparison of “photocatalytic activities” is an inherent problem of the field. The reaction rates calculated per gram of photocatalyst is not suitable for the comparison in a quantitative way since the amount of light scattered and reflected differs usually significantly from experiment to experiment. The book proposed that the practical solution is to compare the optimal rates of various photocatalysts measured in one unique photoreactor at a given lamp intensity. However, it’s really difficult to compare the reaction rates of various reported photocatalysts in one unique photoreactor. Thus, to date, the comparison of product formation rates calculated per gram of various photocatalyst is one of the most common methods to evaluate photocatalytic activities (Nat Commun, 2019, 10, 676; Adv. Optical Mater. 2018, 1800911; Angew. Chem. Int. Ed. 2019, 58, 8103; J. Am. Chem. Soc. 2019, 141, 7615; Angew. Chem. 2019, 131, 17396; Energy Environ. Sci., 2016, 9, 2177; Energy Environ. Sci., 2016, 9, 62; ACS Nano, 2018, 12, 3523; ACS Catal., 2019, 9, 3959; Adv. Energy Mater., 2018, 8, 1701503; Nanoscale Adv., 2019, 1, 4321; Angew. Chem. Int. Ed., 2019, 58, 17236; J. Am. Chem. Soc., 2019, 141, 43, 17431; Small Methods, 2019, 1900586; Adv. Mater., 2019, 31, 1902868; Adv. Mater., 2018, 30, 1706108). Although this comparison may not be rigorous or enough, it has certain significance for the study of enhanced photocatalytic performance of the novel gersiloxenes. Through these comparisons, it is helpful for researchers to understand the correlation between the two-dimensional materials structure and photocatalytic activity, and design highly efficient photocatalysts for a wide range of applications. In addition, we further tested the apparent quantum efficiency (AQE) of the as-synthesized gersiloxene with optimal properties. AQE is considered to be more convincing in proving photocatalytic activity. The gersiloxene with $x = 0.5$ has an AQE as high as 5.95% at 420 nm, which further proves the high activity of the material.

Line 351-354: “In addition, the apparent quantum efficiency (AQE) of gersiloxene with $x=0.5$ measured at different excitation wavelength shows a highest AQE of up to

5.95% at 420 nm (Supplementary Fig. 16), which further indicating the high activity of photocatalyst.”

In supplementary information:

Line 124-146: “*Apparent quantum efficiency (AQE) for photocatalytic CO₂ reduction*”

The photocatalytic activity of the catalyst at a specific excitation wavelength is obtained by a specific band pass filter ($\lambda = 380, 420, 525, 700$ nm) with a 300 W Xe lamp in the same apparatus as the photocatalytic reaction. Generally, the AQE is calculated as follows:

$$AQE(\%) = \frac{\text{number of the electrons taking part in reaction}}{\text{number of incident photons}} \times 100\%$$

For photocatalytic reduction of CO₂ to CO, the AQE is calculated by the following equations,

$$AQE(\%) = \frac{2n(CO)}{I} \times 100\%$$

Where, $n(CO)$ represents the number of molecules of CO production, and I is the number of incident photons. In actual measurement, it is assumed that all incident photons are absorbed by the suspension; the calculation formula of the incident photon number I is as follows:

$$I = PSt \frac{\lambda}{hv}$$

Where, P is the average intensity of the radiation (W/cm²), S is the incident irradiation area (S = 19.63 cm²), t is the irradiation time (t = 4 × 3600 s), λ is the incident wavelength, h is the Planck constant (6.626×10^{-34} J s), and v is the speed of light (3×10^8 m s⁻¹). Therefore, the calculation equation of AQE for photocatalytic reduction of CO₂ to CO can be written as:

$$AQE(\%) = \frac{2CN_A}{PSt \times \frac{\lambda}{hv}} \times 100\%$$

Where, C is the CO production amount (mol); N_A is the Avogadro constant (6.02×10^{23} mol⁻¹).

To further investigate the activity of photocatalytic CO₂ reduction, we performed photocatalytic CO₂ reduction experiments under the monochromatic light irradiations with different wavelength to calculate the apparent quantum efficiency. The test conditions are the same as before except for the irradiations. The photocatalytic reduction of CO₂ to CO activities of gersiloxene with x=0.5 (HGeSiOH) under monochromatic irradiations with wavelengths of 380, 420, 525 and 700 nm are listed in Supplementary Table 4, and the action spectra and wavelength-dependent AQE are shown in Supplementary Fig. 16.

Supplementary Table 4. The data of AQE under incident light with different wavelengths.

Wavelength (nm)	CO production amount (mmol)	P (10 ⁻³ W/cm ²)	AQE (%)
380	7.780	0.41	4.23
420	10.624	0.36	5.95
525	3.492	0.38	1.48
700	1.450	0.19	0.92

Supplementary Fig. 16. The action spectra and wavelength-dependent AQE of gersiloxene with x=0.5 (HGeSiOH).

b) Hydrogen formation in presence of a reducing agent (sodium sulfite) is NOT water splitting, but simple water reduction.

Ans 4): Thank you for your constructive suggestion. We have corrected the wrong description on the photoreduction of water in the manuscript.

Line 30-30: “All gersiloxenes are direct-gap semiconductors and have wide range of light absorption and suitable band positions for light driven water reduction into H₂ and CO₂ reduction to CO under mild conditions.”

Line 103-104: “Tests for water reduction to H₂ showed that the hydrogen production rate is 1.58 mmol g⁻¹ h⁻¹ and that the material has good cycle stability.”

Line 405-407: “Therefore, it can be inferred that in the process of photocatalytic CO₂ or H₂O reduction, CO₂ or H₂O are mainly adsorbed and activated at the -OH site to obtain electrons and further reduced to H₂ and CO.”

Line 423-425: “In addition, the appropriate energy band position endows these materials with photocatalytic properties in H₂O reduction to H₂ and CO₂ reduction to CO.”

Line 433-442: “When used as a photocatalyst in water reduction to produce H₂, it has a hydrogen production rate of 1.58 mmol g⁻¹ h⁻¹ and good cycle stability. When applied to photoreduction of CO₂ under mild conditions (25 °C, 1 atm CO₂), it achieves high conversion efficiency to produce CO with a production rate of 6.91 mmol g⁻¹ h⁻¹ and has a high AQE of 5.95% at 420 nm, which is better than the majority of recently reported photocatalysts. We believe that this research will provide many valuable references for the synthesis, design, regulating electronic properties of new germanium/silicon-based two-dimensional materials and the development of their applications in photocatalysis.”

Lines 244-253:

These are too speculative assumptions. At least at this place of the text.

Ans 5): Thank you for your constructive suggestion. We have revised the manuscript to give a more convincing conclusion. By combining the results of TEM and SEM

with that of N₂ Adsorption/desorption isotherms and pore-size distribution curves analysis, we have come to a more convincing conclusion.

Line 244-262: “The N₂ Adsorption/desorption isotherms and pore-size distribution curves indicate that all gersiloxenes, GeH, and Si₆H₃(OH)₃ have mesoporous structure and gersiloxene with x=0.5 exhibits the most extensive size distribution. Consistent with the differences between SEM images, the surface area continuously increases from 18.9 to 319.7 m²g⁻¹ for the x=0.1 to x=0.5 samples and then drops to 169.4 m²g⁻¹ as the x value increases to 0.9 (Table 1). The TEM results show that the transparency and wrinkles of gersiloxene nanosheets with x=0.5 is the most obvious of all gersiloxenes, indicating that the nanosheets with x=0.5 is the best dispersed, and the interlayer agglomeration effect is the weakest among all samples. Moreover, SEM also showed that the overall size of gersiloxene nanosheets with x=0.5 is significantly smaller than that of other gersiloxenes. These factors lead to the largest specific surface area, which can maximize its contact with the liquid.⁵⁹ It is reported that the large specific surface area may be beneficial to improve the performance of photocatalysts, as it may provide a larger active area that offer paths for the migration of photogenerated carriers, promote the migration of photogenerated electrons and holes between layers, inhibit the recombination of carriers, and boost the release of generated gases.⁶⁰ In contrast, GeH and Si₆H₃(OH)₃ have specific surface areas of 4.7 and 94.8 m²/g, respectively (Supplementary Fig. 7b).”

Supplementary Fig. 7. Adsorption (filled) and desorption (empty) isotherms of N₂ at 77 k for (a) gersiloxenes with x=0.1-0.9; (b) GeH and Si₆H₃(OH)₃. Pore-size distributions of gersiloxenes with x=0.1-0.9 (c), GeH and Si₆H₃(OH)₃ (d).

Line 112-141:

a) How many milligrams of starting materials were used? What was the weight of product obtained?

Ans 6): The weight of starting materials we used are shown in the following Table.

The weight of starting materials (Ca, Ge, Si), intermediate product (CaGe_{2-2x}Si_{2x}), and end product (gersiloxene).

	Ca (mg)	Ge (mg)	Si (mg)	CaGe _{2-2x} Si _{2x} intermediate Product (g)	Gersiloxene Product (mg)
x=0.1	848	2615	112	3.2	600

x=0.3	848	2034	337	2.9	530
x=0.5	848	1453	562	2.4	440
x=0.7	1100	1199	1081	2.8	480
x=0.9	1060	363	1264	2.3	400

Since the final product gersiloxenes is obtained from the upper dispersion obtained by centrifugation after sonication, there is a large amount of product loss.

b) How was product purity proven?

Ans 7): Thank you for the very interesting comment. The issue of purity is significant. In two-dimensional material systems, there is lacking a very accurate method to determine the purity of the product. In our study, to ensure purity. We have done the following. In our study, ensure the purity, we did the following work.

Firstly, the purity of precursor $\text{CaGe}_{2-2x}\text{Si}_{2x}$ is proved by XRD which is always been used to confirm the purity of CaGe_2 and its related Zintl phase alloy materials. (Journal of Solid State Chemistry, 2007, 180, 1575; Acs nano, 2013, 7, 4414; Chem. Mater., 2014, 26, 6941; Journal of Alloys and Compounds, 2019, 774, 502; Journal of Crystal Growth, 2001, 223, 573) which demonstrated that there is almost no elemental Ge and Si or other impurity phases in the product. After the topochemical transformation into gersiloxenes, XRD was first used to further confirm the nonexistence of elemental Ge and Si and unreacted $\text{CaGe}_{2-2x}\text{Si}_{2x}$. Secondly, XPS proved that the zero valence states of elemental germanium and silicon do not exist. Finally, no Ca was detected by EDS characterization, indicating that the CaCl_2 in the product was also washed away. Therefore, we think that the obtained gersiloxenes have high purity and almost no impurities.

c) Are no amorphous by-products formed?

Ans 8): Thank you for the very interesting comment. As described in the manuscript, the equations for topological chemical reactions to synthesize gersiloxenes are as

follows.

Therefore, the by-products of the reaction are mainly H₂ and CaCl₂. H₂ generated during the reaction was taken away by the protective gas argon. After the reaction, CaCl₂ is removed by repeated washing with Milli-Q H₂O and isopropyl alcohol. EDS indicates that there is no calcium in the gersiloxene nanosheets, which demonstrate that CaCl₂ has been completely removed.

d) What is the experimental error in the lattice constant measurements?

Ans 9): Thank you for the question. The lattice constant in the primary manuscript was measured using jade software. In order to more accurately determine the lattice constant of the sample, we reperformed the XRD characterization to obtain the diffraction peak data with ultrahigh intensity, so as to be analyzed using GSAS crystallography data analysis software. The goodness of fit (R factor) is represented by Rietveld R_p and R_{wp} values. The refined lattice parameters and errors are shown in the following table.

Supplementary Table 1. The refined lattice parameters of as-prepared CaGe_{2-2x}Si_{2x}.

x value	0.1	0.3	0.5	0.7	0.9
lattice parameters					
a/b (Å)	3.9837(8)	3.9628(3)	3.9268(1)	3.8927(6)	3.8613(4)
c (Å)	30.6151(8)	30.5844(7)	30.5792(6)	30.5224(1)	30.5763(2)
Rwp	9.94%	8.78%	4.59%	6.19%	8.82%
Rp	7.32%	6.12%	2.99%	3.87%	5.95%

Line 122-124: “Moreover, the lattice constant a gradually changes from 3.9837 to 3.8613 Å as x increases from 0.1 to 0.9 (shown in Supplementary Table 1), following

the Vegard's law”

Lines 243-244:

Giving two decimal places for the specific surface seems a bit overdone

Ans 10): Thank you for your constructive suggestion. We have revised the decimal places in the corresponding positions of the manuscript.

Lines 247-250: “Consistent with the differences between SEM images, the surface area continuously increases from 18.9 to 319.7 m²g⁻¹ for the x=0.1 to x=0.5 samples and then drops to 169.4 m²g⁻¹ as the x value increases to 0.9 (Table 1),”

Lines 260-262: “In contrast, GeH and Si₆H₃(OH)₃ have specific surface areas of 4.7 and 94.8 m²g⁻¹, respectively (Supplementary Fig. 7b).”

Table 1. Summary of the Ge/Si ratios and BET surface areas of 2D gersiloxenes with varying x values.

x	0.1	0.3	0.5	0.7	0.9
Theoretical Ge/Si ratio	9:1	7:3	1:1	3:7	1:9
Experimental Ge/Si ratio detected by EDS	8.44:1	8.89:3	1:1.07	3:6.36	1:8.77
BET surface Area (m ² g ⁻¹)	18.9	46.2	319.7	228.7	169.4

Lines 247-253:

Here are unnecessary, too speculative assumptions made. At least at this place of the text.

Ans 11): Thank you for your constructive suggestion. We have revised the manuscript to give a more convincing conclusion. By combining the results of TEM and SEM with that of N₂ Adsorption/desorption isotherms and pore-size distribution curves analysis, we have come to a more convincing conclusion.

Line 244-262: “The N₂ Adsorption/desorption isotherms and pore-size distribution curves indicate that all gersiloxenes, GeH, and Si₆H₃(OH)₃ have mesoporous structure and gersiloxene with x=0.5 exhibits the most extensive size distribution. Consistent with the differences between SEM images, the surface area continuously increases from 18.9 to 319.7 m²g⁻¹ for the x=0.1 to x=0.5 samples and then drops to 169.4 m²g⁻¹ as the x value increases to 0.9 (Table 1). The TEM results show that the transparency and wrinkles of gersiloxene nanosheets with x=0.5 is the most obvious of all gersiloxenes, indicating that the nanosheets with x=0.5 is the best dispersed, and the interlayer agglomeration effect is the weakest among all samples. Moreover, SEM also showed that the overall size of gersiloxene nanosheets with x=0.5 is significantly smaller than that of other gersiloxenes. These factors lead to the largest specific surface area, which can maximize its contact with the liquid.⁵⁹ It is reported that the large specific surface area may be beneficial to improve the performance of photocatalysts, as it may provide a larger active area that offer paths for the migration of photogenerated carriers, promote the migration of photogenerated electrons and holes between layers, inhibit the recombination of carriers, and boost the release of generated gases.⁶⁰ In contrast, GeH and Si₆H₃(OH)₃ have specific surface areas of 4.7 and 94.8 m²/g, respectively (Supplementary Fig. 7b).”

Supplementary Fig. 7. Adsorption (filled) and desorption (empty) isotherms of N₂ at 77 k for (a) gersiloxenes with x=0.1-0.9; (b) GeH and Si₆H₃(OH)₃. Pore-size distributions of gersiloxenes with x=0.1-0.9 (c), GeH and Si₆H₃(OH)₃ (d).

Lines 257-270:

a) Discussion of the DRS is too superficial: almost all samples (especially the one with x=0.5) exhibit pronounced sub-bandgap absorptions which may be of crucial influence on the photocatalytic activity.

Ans 12): Thank you for the constructive suggestion.

The sub band gap absorption phenomenon of semiconductor materials is common in the UV spectrum test, which has a certain impact on the photocatalytic performance. We have carried out an in-depth discussion of the UV-Vis spectra combined with the results of XPS characterization. In addition, the BET, PL, ESR and photoelectrochemical test were used to explain the photocatalytic activities.

Line 212-231: “The high-resolution Ge3d XPS spectra exhibit peaks at ca. 30 eV, corresponding to the Ge-Ge bonds (Fig. 4f), and the shoulder peaks around 32.7eV are ascribed to Ge-O bonds, which is due to the trace oxidation on the surface similar to that of germanium nanosheets.⁵⁵⁻⁵⁶ In the Si2p spectra, the peaks were located at binding energies of approximately 99.9 and 103.0 eV, corresponding to Si-Si bonds and Si-O bonds in the two-dimensional Si chain network (Fig. 4e), respectively. By contrast, the Ge-Ge and Si-Si bonds for the precursor $\text{CaGe}_{2-2x}\text{Si}_{2x}$ are located at 28.8 and 98.5 eV (Supplementary Fig. 4), respectively. This result indicates that $\text{CaGe}_{2-2x}\text{Si}_{2x}$ transforms to 2D $\text{Ge}_{1-x}\text{Si}_x\text{H}_{1-y}(\text{OH})_y$. In addition, the Si2p spectra show that the content of the Si-Si bond relative to that of the Si-O bond in $\text{Ge}_{1-x}\text{Si}_x\text{H}_{1-y}(\text{OH})_y$ increases with an increasing x (when $x=0.9$, the content of the Si-Si bond is obviously higher than that of the Si-O bond), further indicating that when $x > 0.5$, each Si atom is terminated not only by OH but also by H. Moreover, O1s spectrum in the range of 527 eV~536 eV exhibits the states of surface oxygen on samples (Fig. 4d). The O1s peaks can be also fitted into several components. Peaks at around 531, 532.4, and 533.1 eV are attributed to the Ge-O, Si-O and O-H, respectively. For GeH, $\text{Si}_6\text{H}_3(\text{OH})_3$ and all gersiloxenes, a same fitted peak located at around 531.7 eV, is supposed to be associated with the surface oxygen vacancies,⁵⁷⁻⁵⁸ which would significantly influence their optical properties.”

Fig. 4. (a) FTIR and (b) Raman spectra of 2D $\text{Ge}_{1-x}\text{Si}_x$ alloys with different x values. (c-f) XPS spectra of 2D $\text{Ge}_{1-x}\text{Si}_x\text{H}_{1-y}(\text{OH})_y$ ($x=0.1, 0.3, 0.5, 0.7, 0.9$), GeH and $\text{Si}_6\text{H}_3(\text{OH})_3$. (c) XPS survey spectra. High-resolution XPS spectra of (d) O1s (e) Si2p, and (f) Ge3d.

Line 266-287: “As shown in Fig. 5a, all gersiloxene samples have a wide range of

light absorption from ultraviolet to visible light. The absorption range of gersiloxenes is between that of GeH and $\text{Si}_6\text{H}_3(\text{OH})_3$ (supplementary information Fig. 13a) With the increasing x, the absorption edge shifts to the direction of short wavelength, which is consistent with the change in sample's color from dark red, brick red, brown, yellow green, to light green (Fig. 5a inset). It is worth noting that gersiloxene shows obvious sub-bandgap absorptions compared with pure GeH and pure $\text{Si}_6\text{H}_3(\text{OH})_3$. This sub-bandgap absorption is supposed to be caused by the oxygen vacancies, which is indicated by the XPS O1s spectra. Gersiloxene with $x = 0.5$ exhibits the most significant sub-bandgap absorption, indicating that it has the highest oxygen vacancies concentration than other ones which may be attributed to its graphene-like 2D structure and the highest specific surface area among the as-synthesized gersiloxenes. The significant wide sub-bandgap absorption further increases the light absorption range of the material, which is advantageous for the enhanced photocatalytic and photoelectrochemical performance. Fig. 5b presents the corresponding Tauc plots, which are calculated based on the assumption that the gersiloxenes are direct-bandgap semiconductor materials and proved by subsequent theoretical calculations. As shown in Fig. 5b, with an increasing x, the bandgap increased from 1.8 eV ($x=0.1$) to 2.57 eV ($x=0.9$), indicating that different x values (the content of Si) endowed the 2D gersiloxenes with different bandgap and a wide adjustable absorption range that can be linearly regulated.”

Line 355-377: “The photogenerated carrier recombination and separation behaviors of 2D gersiloxenes were further investigated based on PL spectroscopy. It can be seen from Fig. 6g that all gersiloxenes have a wide range of fluorescence emissions in the range of 440-650 nm. The larger portion of emission signals at 440-530 nm are ascribed to the free exciton from electron–hole recombination. While the peak at 550-600 nm corresponded to the recombination of two \square electron \square trapped oxygen vacancies with photogenerated holes. In addition, the PL intensity shows an obvious decrease from $x=0.1$ to $x=0.5$ and a subsequent increase from $x=0.5$ to $x=0.9$. For

comparison, the PL intensity of GeH and $\text{Si}_6\text{H}_3(\text{OH})_3$ (Supplementary Fig. 14a) were also tested, both of which are higher than that of the gersiloxene with $x=0.5$. These results demonstrate that the gersiloxene with $x=0.5$ has better capability to effectively suppress the carrier recombination than other gersiloxenes, GeH and $\text{Si}_6\text{H}_3(\text{OH})_3$. Thus, the extensive light absorption ability together with the less radiative electron–hole recombination endowed the gersiloxene ($x=0.5$, HGeSiOH) with significantly enhanced photoactivity and photoelectrochemical performance. A photoelectrochemical test further confirmed that HGeSiOH shows the best photocurrent performance and the lowest interfacial resistance of charge carriers (Fig. 6h and I and Supplementary Fig. 14b and c), coinciding with the HER and COER results. As HGeSiOH has the highest surface area of $319.7 \text{ m}^2 \text{ g}^{-1}$, the distinctive two-dimensional nanostructure and large specific surface area of HGeSiOH provide more paths for the migration of photogenerated carriers, promote the migration of photogenerated electrons and holes between layers, and inhibit the recombination of carriers, thus greatly improving the photocatalytic performance of materials.”

Fig. 6. Time-dependent photocatalytic hydrogen evolution (a) and HERs (b) of gersiloxenes with $x=0.1-0.9$. (c) Photostability for H₂ production of the gersiloxene with $x=0.5$ (HGeSiOH). Time-dependent photocatalytic CO evolution (d) and COERs (e) of gersiloxenes with $x=0.1-0.9$. (f) Time-dependent photocatalytic CO evolution for 10 h. PL spectra (g), EIS Nyquist plots (h) and transient photocurrent responses (300 W xenon lamp) (i) of gersiloxenes with $x=0.1-0.9$.

Supplementary Fig. 14 PL spectra (a), EIS Nyquist plots (b) and transient photocurrent responses (300 W xenon lamp) (c) of GeH, $\text{Si}_6\text{H}_3(\text{OH})_3$ and HGeSiOH. (d) HER and COER of GeH and $\text{Si}_6\text{H}_3(\text{OH})_3$.

Line 408-416: “We also performed an electron spin resonance (ESR) test on the sample. ESR spectra indicate that HGeSiOH has abundant oxygen vacancies and changes little before and after illumination (This further explains the results of oxygen vacancies in XPS, UV and the PL characterization). These holes may be caused by either Ge or Si atoms because oxygen vacancies are also present in GeH (caused by oxidation) and $\text{Si}_6\text{H}_3(\text{OH})_3$ (Supplementary Fig. 17). Obviously, the concentration of oxygen vacancies in HGeSiOH is higher than that of GeH and $\text{Si}_6\text{H}_3(\text{OH})_3$. The presence of surface oxygen vacancy defects leads to electron enrichment, which enhances the activation of CO_2 molecules, thereby further promoting CO_2 reduction⁶⁵⁻⁶⁷.”

Supplementary Fig. 17. ESR spectra of GeH, HGeSiOH, and $\text{Si}_6\text{H}_3(\text{OH})_3$ under dark (a) and light (b).

b) Bandgaps obtained from the depicted Tauc plots should have a rather large experimental error. What was the reproducibility?

Ans 12): Thank you for the very interesting comment.

To verify the reproducibility, we performed five replicate experiments on each ratio of gersiloxene. The measured Tauc plots, the bandgap values and average value for each gersiloxene are shown in the following figure and table. It can be seen that for the gersiloxene of each ratio, the Tauc curves of the five replicate experiments are relatively close, and the band gap values are slightly agitated within a reasonable range. This proved the reproducibility of the experiment.

Tauc plots of 2D $\text{Ge}_{1-x}\text{Si}_x\text{H}_{1-y}(\text{OH})_y$ from five experiments for each x value. (a) $x=0.1$, (b) $x=0.3$, (c) $x=0.5$, (d) $x=0.7$, (e) $x=0.9$.

Bandgap values of five replicate experiments for each gersiloxene.

	Sample 1	Sample 2	Sample 3	Sample 4	Sample 5	average value
$x=0.1$	1.81	1.84	1.83	1.83	1.80	1.82
$x=0.3$	2.00	1.94	1.95	1.96	1.91	1.95
$x=0.5$	2.30	2.40	2.42	2.40	2.43	2.39
$x=0.7$	2.53	2.48	2.55	2.54	2.53	2.53
$x=0.9$	2.57	2.66	2.61	2.65	2.63	2.62

c) Comparison of weight-related rection rates is not reliable (see first remark above)

Ans 13): Thank you for the very interesting comment.

As mentioned before, the comparison of “photocatalytic activities” is an inherent problem of the field, which needs the joint efforts of researchers in the whole industry to solve. Therefore, at present, comparing product formation rates calculated per gram

of various photocatalyst is the most common method to compare photocatalytic activities (Nat Commun, 2019, 10, 676; Adv. Optical Mater. 2018, 1800911; Angew. Chem. Int. Ed. 2019, 58, 8103; J. Am. Chem. Soc. 2019, 141, 7615; Angew. Chem. 2019, 131, 17396; Energy Environ. Sci., 2016, 9, 2177; Energy Environ. Sci., 2016, 9, 62; ACS Nano, 2018, 12, 3523; ACS Catal., 2019, 9, 3959; Adv. Energy Mater., 2018, 8, 1701503; Nanoscale Adv., 2019, 1, 4321; Angew. Chem. Int. Ed., 2019, 58, 17236; J. Am. Chem. Soc., 2019, 141, 43, 17431; Small Methods, 2019, 1900586; Adv. Mater., 2019, 31, 1902868; Adv. Mater., 2018, 30, 1706108). Although this comparison may not be rigorous or accurate enough, it has certain significance for the study of new photocatalytic materials. Through these comparisons, it may have a better reference value for potential researchers to select appropriate materials for in-depth study. In addition, we further tested the apparent quantum efficiency (AQE) of the as-synthesized gersiloxene with optimal properties. AQE is considered to be more convincing for the characterization of photocatalytic activity. The gersiloxene with $x = 0.5$ has an AQE as high as 5.95% at 420 nm, which further proves the high activity of the material.

Reviewer #2 (Remarks to the Author):

This work reports a new 2D GeSi alloy with enhanced photocatalytic performance for H₂ evolution and CO₂ photoreduction to CO. By means of topotactic deintercalation process the authors synthesized a two-dimensional honeycomb-like Ge-Si alloy compounds terminated by -H/-OH (gersiloxenes). The experimental results and theoretical calculations confirmed the gersiloxenes are all direct-bandgap semiconductors with monotonical increase of bandgap (1.8-2.57 eV) as Si content x increase ($x = 0.1-0.9$). Among the synthesized gersiloxene the HGeSiOH ($x=0.5$) has highest specific surface area (319.7 m² g⁻¹) and shows excellent photocatalytic capabilities, providing a new possible option in the field of catalysts for H₂ evolution and CO₂ photoreduction to CO.

The paper presents the detailed characterizations of synthesized samples to measure various physical and chemical properties, electrochemical and photocatalytic data. The theoretical calculations of DFT for band structures and adsorption energy of H₂O and CO₂ on HGeSiOH, Si₆H₃(OH)₃ and GeH have been conducted to understand in depth the photocatalytic behaviors of samples. Some discussions are as followed.

1. In the manuscript “The theoretical calculations (Supplementary Fig. 8 and 9) proved that the gersiloxenes are all direct bandgap semiconductors. And the calculated bandgap values agree well with the experimentations.” Pick the experimental bandgap data in the manuscript and calculational bandgap data in the supplementary, draw them in the chart as following. (please find the chart in the attachment - comment for author) The trends are similar except the case at x=0.3. Please check or calculate again the data at x=0.3 carefully. Another question: what is the authors’ purpose to calculate the two layers (bulk) model, which is not discussed in the manuscript/ supplementary.

Ans 1): Thank you for your constructive suggestion.

We have carefully checked all the calculation process and found that there are errors with the parameter setting in the calculation of bulk structure at x = 0.3, resulting in the deviation of the calculation results. After correction, we recalculated the data at x=0.3 carefully, and we have revised the corresponding content of the manuscript. After computational optimization, the new energy band structure and PDOS are obtained by further calculations. And we have drawn the calculated and experimental bandgap data in the chart. The recalculated bandgap of bulk gersiloxene with x=0.3 is 1.66 eV, which is in accordance with the trends of all gersiloxene structures. Because the as-prepared gersiloxene nanosheets have few-layer structures with thickness of about 3-6 nm. The computational calculation results show that the obtained gersiloxene nanosheets are 2D materials with the properties of monolayer nanostructures.

Supplementary Fig 11. Evolution curves of band gap value depending on x for the experiment, theoretical calculation results with bulk and monolayer structures.

On the other hand, we provide additional explanations in supporting information for calculating the two layers (bulk) model. Calculating bulk and single-layer structures for novel two-dimensional materials is a common practice, which helps to comprehensive understand the electronic structural properties of new two-dimensional materials. (Acs nano, 2013, 7, 4414; Nanoscale, 2018,10, 15989; Chem. Mater. 2017, 29, 6261; Appl. Surf. Sci., 2019, 467–468, 881–888) Through this calculation, we can determine the experimental properties of the materials. As is known to all, 2D materials have a trend of re-stacking after the liquid phase exfoliation into monolayer nanosheets. Whereas the as-prepared gersiloxene nanosheets in our experiment have a thickness of about 3-6 nm, which belongs to a few-layer structure and is stacked by several monolayer nanosheets. Our calculation results show that the obtained gersiloxene nanosheets are 2D materials with the properties of monolayer nanostructures, rather than forming layered stacked bulk structures under the influence of van der Waals forces. Moreover, the calculated results demonstrate that gersiloxenes are all direct-gap semiconductors regardless of the proportions of Ge and Si or the bulk and monolayer structure.

In supplementary information

Line 271-274: “In addition, bulk structures show the same bandgap type as monolayer structures; that is, they are all direct bandgap semiconductors, and the bandgap values are 1.58, 1.66, 1.82, 1.96, and 2.05 eV, respectively (Supplementary Fig. 10)”

In supplementary information

Line 282-292: “Since the gersiloxene nanosheets obtained in our experiment are prepared from the liquid phase dispersion, which are few-layer stacking structure with a thickness of about 3-6nm. As is known to all, 2D materials have a trend of re-stacking after the liquid phase exfoliation into monolayer nanosheets. Therefore, the calculated results help us to judge whether the properties of the obtained few-layer nanosheets are closer to that of the single-layer structure or the bulk structure. It can help us understand theoretical calculations and experimental results. Obviously, the electronic properties of the experimentally obtained few-layer nanosheets are more in conformity with the single-layer structure, rather than the bulk structure that is tightly packed by the layers under the van der Waals force (Supplementary Fig. 11).”

2. Just above the conclusion in the manuscript “These holes may be caused by either Ge or Si atoms because oxygen vacancies are also present in GeH and Si6H3(OH)3 (Supplementary Fig. 12).” How to understand that the oxygen vacancies present in GeH? Please provide clear description.

Ans 2): Thank you for your constructive suggestion.

The oxygen vacancies presented in GeH are caused by the surface oxidation of GeH in air. It is reported that GeH will be trace oxidized in the air to form GeOx (Acs nano, 2013, 7, 4414, Catal. Commun., 2019, 11846), whereas O vacancies would form during the oxidization process. (Appl. Phys. Lett., 2009, 94, 142903) It has been demonstrated by the XPS O1s spectra in the revised manuscript.

Line 212-231: “The high-resolution Ge3d XPS spectra exhibit peaks at ca. 30 eV, corresponding to the Ge-Ge bonds (Fig. 4f), and the shoulder peaks around 32.7eV are ascribed to Ge-O bonds, which is due to the trace oxidation on the surface similar to that of germanium nanosheets.⁵⁵⁻⁵⁶ In the Si2p spectra, the peaks were located at

binding energies of approximately 99.9 and 103.0 eV, corresponding to Si-Si bonds and Si-O bonds in the two-dimensional Si chain network (Fig. 4e), respectively. By contrast, the Ge-Ge and Si-Si bonds for the precursor $\text{CaGe}_{2-2x}\text{Si}_{2x}$ are located at 28.8 and 98.5 eV (Supplementary Fig. 4), respectively. This result indicates that $\text{CaGe}_{2-2x}\text{Si}_{2x}$ transforms to 2D $\text{Ge}_{1-x}\text{Si}_x\text{H}_{1-y}(\text{OH})_y$. In addition, the Si2p spectra show that the content of the Si-Si bond relative to that of the Si-O bond in $\text{Ge}_{1-x}\text{Si}_x\text{H}_{1-y}(\text{OH})_y$ increases with an increasing x (when $x=0.9$, the content of the Si-Si bond is obviously higher than that of the Si-O bond), further indicating that when $x > 0.5$, each Si atom is terminated not only by OH but also by H. Moreover, O1s spectrum in the range of 527 eV~536 eV exhibits the states of surface oxygen on samples (Fig. 4d). The O1s peaks can be also fitted into several components. Peaks at around 531, 532.4, and 533.1 eV are attributed to the Ge-O, Si-O and O-H, respectively. For GeH, $\text{Si}_6\text{H}_3(\text{OH})_3$ and all gersiloxenes, a same fitted peak located at around 531.7 eV, is supposed to be associated with the surface oxygen vacancies,⁵⁷⁻⁵⁸ which would significantly influence their optical properties.”

Line 408-416: “We also performed an electron spin resonance (ESR) test on the sample. ESR spectra indicate that the material has abundant oxygen vacancies and does not change much before and after illumination (This further explains the results of oxygen vacancies in XPS, UV and the PL characterization). These holes may be caused by either Ge or Si atoms because oxygen vacancies are also present in GeH (caused by oxidation) and $\text{Si}_6\text{H}_3(\text{OH})_3$ (Supplementary Fig. 17).”

3. In this work the calculations of band structure are based on VASP, while the calculations of adsorption energy are based on Dmol3 module in Materials Studio software. How about directly use VASP to perform the calculations of adsorption energy (which material researchers often do)? What is the authors' consideration? Comparison or discussion would be helpful for the interest of relevant researchers.

Ans 3): Thank you for your constructive suggestion.

We have performed the calculations of adsorption energy using VASP. The calculated results are as follows.

In supplementary information

Line 179-182: “Moreover, for comparison, we used VASP to perform the calculations of adsorption energy using the same parameter as the Dmol3 module in Materials Studio software, this would be helpful for the interest of relevant researchers.”

Supplementary Fig. 18 VASP optimized geometric structures and binding energies for H₂O (a) and CO₂ (b) adsorption on HGeSiOH, Si₆H₃(OH)₃, and GeH monolayers.

In supplementary information

Line 382-393: “In order to better understand the influence of structure on adsorption energy, we also used VASP to calculate the adsorption energy for comparison which material researchers often do. As shown in Supplementary Fig. 18a and b, the H₂O or CO₂ adsorption energy on HGeSiOH is more negative than that on Si₆H₃(OH)₃ and GeH, suggesting that the alloyed structure of 2D GeSi can significantly improve its

H₂O or CO₂ adsorption capacity. The difference from the result of Materials Studio calculation is the VASP calculations demonstrated that the interactions of an H₂O or CO₂ molecule with the Ge-H and Si-H of HGeSiOH and Si₆H₃(OH)₃ are not energetically favorable, which indicates that Si-OH in HGeSiOH and Si₆H₃(OH)₃ has a strong adsorption effect on H₂O and CO₂, which makes the adsorption of Ge-H for H₂O and CO₂ unstable. Therefore, it can be concluded that both H₂O and CO₂ may be adsorbed and activated at the -OH sites.”

Reviewer #3 (Remarks to the Author):

This report by Zhao et al. entitled “Novel two-dimensional gersiloxenes with tunable bandgap for photocatalytic H₂ evolution and CO₂ photoreduction to CO” synthesizes 2D -H/-OH terminal substituted siligenes, which they term as gersiloxnes. These are made from CaGe₂-2xSi₂x. The 2D materials are characterized by XRD, TEM, AFM, FTIR, Raman, XPS, UV-Vis diffuse reflectance spectroscopy. The authors show photocatalytic activity for H₂ and CO generation. Furthermore, the authors rely on DFT calculations to determine the electronic band structure. This is an interesting material system that seems to be promising and would be of wide interest. I recommend that this paper be reconsidered after major edits.

First, the figures are very small and I am not confident in the author’s analysis because the figure size is too small. Also, the color choices and line widths need to be selected such that they can be easily interpreted. XPS data should be plotted from high to low energy, which is the convention for XPS.

Ans 1): Thank you for your constructive suggestion. We have adjusted the figures to present the convincing results. In particular, in order to show the results more clearly and accurately, we reperformed the XRD characterization to obtain the diffraction peak data with higher intensity, so as to draw a higher identification figure to present the results of our analysis. And the data of the control samples of x=0 and x=1, i.e.,

precursor CaGe_2 and CaSi_2 , and the corresponding topotactic deintercalation products GeH and $\text{Si}_6\text{H}_3(\text{OH})_3$, are added to the figures to give confidence in the peak assignments. XPS data have been plotted from high to low energy.

Fig. 1. (a) Schematic illustration of topotactic deintercalation of $\text{CaGe}_{2-2x}\text{Si}_{2x}$ to $(\text{GeH})_{1-x}(\text{SiOH})_x$ ($x < 0.5$) or $(\text{GeH})_{1-x}\text{Si}_x(\text{OH})_{0.5}\text{H}_{x-0.5}$ ($x \geq 0.5$) (Ca, blue; Ge, green; H, white; O, red; Si, yellow). (b) XRD patterns of the precursor $\text{CaGe}_{2-2x}\text{Si}_{2x}$ ($x=0, 0.1, 0.3, 0.5, 0.7, 0.9, 1$), CaGe_2 and CaSi_2 . The vertical lines at the top and bottom are the Inorganic Crystal Structure Database (ICSD) for CaSi_2 (ICSD154431) and CaGe_2 (ICSD245612). (c) XRD patterns of the topotactic deintercalation products $\text{Ge}_{1-x}\text{Si}_x\text{H}_{1-y}(\text{OH})_y$ ($x=0, 0.1, 0.3, 0.5, 0.7, 0.9, 1$), GeH and $\text{Si}_6\text{H}_3(\text{OH})_3$.

Fig. 4. (a) FTIR and (b) Raman spectra of 2D $\text{Ge}_{1-x}\text{Si}_x$ alloys with different x values. (c-f) XPS spectra of 2D $\text{Ge}_{1-x}\text{Si}_x\text{H}_{1-y}(\text{OH})_y$ ($x=0.1, 0.3, 0.5, 0.7, 0.9$), GeH and $\text{Si}_6\text{H}_3(\text{OH})_3$. (c) XPS survey spectra. High-resolution XPS spectra of (d) O1s (e) Si2p, and (f) Ge3d.

In general, I think that the authors did all of the correct characterization methods and catalysis tests; however, I am not confident in the results. For the XPS data provided in the SI, there appears to be intensity at 0eV for all of the spectra shown in the VB region. Also, the figures should be plotted such that this VB area (-1 to 5 eV) is shown. This would give reviewers confidence (or not) in the fits, which directly provides the data for Figure 5c.

Ans 2): Thank you for your constructive suggestion. We have adjusted the figures to give the convincing results. The figures have been plotted in the range of -1-5 eV.

Supplementary Fig. 12. XPS valence band spectra of 2D gersiloxenes with different x value. (a) 0.1, (b) 0.3, (c) 0.5, (d) 0.7, (e) 0.9.

Supplementary Fig. 13 (a) UV-vis diffuse reflectance spectra and Tauc plots (b) of GeH and Si₆H₃(OH)₃. (c) XPS valence band spectra of GeH and Si₆H₃(OH)₃. (d) Energy band structure of GeH and Si₆H₃(OH)₃.

Another example is the photocatalytic testing. The controls should be more clearly discussed with the results shown in SI. No light, no catalyst and in the presence of N₂ gas for the CO₂ photoreduction.

Ans 3): Thank you for your constructive suggestion. We have drawn a figure to illustrate the results of controlling tests, and the discussion with the results are shown in the Supplementary information.

Supplementary Fig. 15 Photocatalytic CO₂ reduction rates of HGeSiOH with purging highly-pure N₂ instead of CO₂, no light, and no catalyst.

In supplementary information

Line 350-357: “Comparison tests were performed to confirm the photocatalysis ability of the synthesized catalyst. The results are shown in Supplementary Fig. 15. To exclude the possibility of influence of organic impurities on the sample surface, highly-pure N₂ is purged into the reactor instead of CO₂. No CO can be detected during the photoreaction process in N₂ which proves that CO was not produced by the possible organic impurities on the gersiloxene sample surface. Moreover, no CO was detected with no light irradiation or without gersiloxene catalyst, further confirming that it is the gersiloxene catalyst initiating the reaction.”

Control samples of x=0 and x=1 should also be shown for the various characterization methods. This would give confidence in the peak assignments.

Ans 4): Thank you for your constructive suggestion.

We have added various characterization for control samples of x=0 and x=1.

Fig. 1. (a) Schematic illustration of topotactic deintercalation of $\text{CaGe}_{2-2x}\text{Si}_{2x}$ to $(\text{GeH})_{1-x}(\text{SiOH})_x$ ($x < 0.5$) or $(\text{GeH})_{1-x}\text{Si}_x(\text{OH})_{0.5}\text{H}_{x-0.5}$ ($x \geq 0.5$) (Ca, blue; Ge, green; H, white; O, red; Si, yellow). (b) XRD patterns of the precursor $\text{CaGe}_{2-2x}\text{Si}_{2x}$ ($x=0, 0.1, 0.3, 0.5, 0.7, 0.9, 1$), CaGe_2 and CaSi_2 . The vertical lines at the top and bottom are the Inorganic Crystal Structure Database (ICSD) for CaSi_2 (ICSD154431) and CaGe_2 (ICSD245612). (c) XRD patterns of the topotactic deintercalation products $\text{Ge}_{1-x}\text{Si}_x\text{H}_{1-y}(\text{OH})_y$ ($x=0, 0.1, 0.3, 0.5, 0.7, 0.9, 1$), GeH and $\text{Si}_6\text{H}_3(\text{OH})_3$.

Fig. 4. (a) FTIR and (b) Raman spectra of 2D $\text{Ge}_{1-x}\text{Si}_x$ alloys with different x values. (c-f) XPS spectra of 2D $\text{Ge}_{1-x}\text{Si}_x\text{H}_{1-y}(\text{OH})_y$ ($x=0.1, 0.3, 0.5, 0.7, 0.9$), GeH and $\text{Si}_6\text{H}_3(\text{OH})_3$. (c) XPS survey spectra. High-resolution XPS spectra of (d) O1s (e) Si2p, and (f) Ge3d.

Supplementary Fig. 14 PL spectra (a), EIS Nyquist plots (b) and transient photocurrent responses (300 W xenon lamp) (c) of GeH, $\text{Si}_6\text{H}_3(\text{OH})_3$ and HGeSiOH . (d) HER and COER of GeH and $\text{Si}_6\text{H}_3(\text{OH})_3$.

Figure 4c shows that the samples have Cl. Cl is a great passivation layer. How do the authors know that the surfaces don't have -Cl and only have -O/-OH? I would imagine Cl is present and important.

Ans 5): Thank you for your great question.

Trace Ge-Cl and Si-Cl often generate in the topotactic deintercalation process of Zintl-phase CaGe_2 and CaSi_2 , therefore, chlorine was also detected in the synthesized gersiloxenes by XPS. However, for full-hydrogenated germanene, the electronic structure will change significantly when the Cl substitution of chlorine exceed 25% (Comput. Mater. Sci., 2014, 92, 244). Whereas the effect of low chlorine content on electronic properties of Ge or Si based graphene-like materials is negligible. As is been demonstrated in literatures (Acs nano, 2013, 7, 4414; Chem. Mater., 2014, 26, 6941; Chem. Commun., 2014, 50, 11046; Jpn. J. Appl. Phys., 1996, 35, L293; Mater.

Res. Bull., 1996, 31, 307.) From the xps results, we obtained that the chlorine in the molar ratio Cl/(Ge+Si) are 0.05, 0.04, 0.06, 0.04, and 0.05 for gersiloxenes with x=0.1, 0.3, 0.5, 0.7, and 0.9, respectively. The content of Cl atoms is very low, so the influence on the electronic properties of the gersiloxenes is negligible.

Also, with Figure 5c, how did the authors determine the energetics for the H+ and CO2 reduction. This should be discussed in the text. Moreover, XPS is performed under ultra-high vacuum and H+ and CO2 reduction is in electrolyte. How are you able to be put on the same energy scale. This should be discussed.

Ans 6): Thank you for your insightful comments.

The reaction energy levels for the transformation of CO₂ into CO and water reduction are known and commonly used data, which can be taken from the literature. (Energy Environ. Sci., 2016, 9, 2177; Small Methods 2017, 1, 1700080; Energy Environ. Sci. 2010, 3,43–81.) We have revised the manuscript to clarify it.

Line 309-311: “The resulting electronic band structures are shown in Fig. 5c (The reaction energy levels for the transformation of CO₂ into CO and water reduction (H⁺/H₂) and oxidation (O₂/H₂O) are taken from the literature⁶¹).”

For the other question, the difference does exist between the XPS measurement condition and the actual working condition of the material. XPS measurement is performed under ultra-high vacuum (UHV), and the actual working condition of the catalysts is usually the atmospheric condition. The electronic structure information derived on catalysts in XPS was under UHV conditions, and correlation among them could only be established for limited system. The pressure difference between UHV measurement conditions in XPS and the actual working conditions of the materials is known as “pressure gap” in surface science. (J. Phys. Chem. B, 2005, 109, 13272; Catal. Lett. 2007, 119, 50; J. Phys. Chem. C, 2013, 117, 4717) However, the short mean-free path of electrons with energies below 1500 eV in a gas at ambient pressure does not allow XPS analyses to be performed under realistic experimental conditions, from an applications point of view, and a vacuum level is required for X-ray anodes

and channeltrons. (Rev. Sci. Instrum., 2012, 83, 093112; Surf. Sci. Rep., 2008, 63, 169) These problems can only be partly overcome by performing near ambient pressure XPS (NAXPS). This technique relies on the use of differential pumping stages, the minimisation of the specimen–aperture distance in the high-pressure regions, and the addition of electrostatic lenses to focus the electrons through the differential pumping scheme. Among the most effective systems available, the pressure in the analysis chamber (AC) can reach tens of mbar, with acceptable photoelectron intensity losses. But these techniques suffer from many drawbacks, such as reduced energy resolution and specimen freedom of movement, high cost, and accessibility to the facilities, as most of them are dedicated for operation at synchrotron light sources. Unfortunately, it has to be concluded that the so-called “pressure gap” in XPS can still only be partially filled with a limited maximum pressure, lower spectrometer resolution, and high cost. (Surf. Sci. Rep., 2008, 63, 169; Relat. Phenom., 2010, 177, 71; Rev. Sci. Instrum., 2010, 81, 053106; Chem. Commun., 2012, 48, 3812; Surf. Interface Anal., 2012, 44, 1100; ChemSusChem, 2019, 12, 621)

It is known to all that the electronic structure of a semiconductor material is mainly composed of a valence band and a conduction band, and there is a certain gap between the valence band and the conduction band. The difference between the conduction band minimum and the valence band maximum is the semiconductor bandgap. When the semiconductor photocatalysts are irradiated by the light with energy equal or greater than the bandgap of themselves, the electrons in the photocatalyst crystal are excited from the valence band to the conduction band, generate free electrons and holes at the conduction band and valence band and transfer from the inside of the semiconductor photocatalysts to the surface. The photogenerated electrons and holes exhibit strong reducing ability and strong oxidizing ability, respectively. As long as the bandgap energy of the synthesized photocatalyst match the photoenergy, the position of the conduction band and the valence band match the reaction energy levels for the corresponding reduction and oxidation of target reactant (such as CO₂ and H₂O), the photocatalyst can be used to

drive the photooxidation and photoreduction to obtain the corresponding products. Therefore, determining the bandgap and the position of the conduction band and valence band of the synthesized semiconductor material is very important for judging whether the material can be used for photocatalytic reaction. To date, valence-band XPS measurement is one of the most commonly used methods to assess the valence band of a semiconductor material. (Nature catalysis, 2018, 1, 704; J. Am. Chem. Soc. 2018, 140, 9078; Energy Environ. Sci., 2014, 7, 1700; Angew. Chem. Int. Ed. 2019, 58, 8103; J. Am. Chem. Soc. 2019, 141, 4209; Applied Catalysis B: Environmental, 2019, 242, 312; Adv. Energy Mater. 2018, 8, 1801084). XPS measurement is not the factor that affects the valence band of semiconductor materials, therefore, the measurement of XPS under ultra-high vacuum is not contradictory to the measurement of photocatalytic H⁺ and CO₂ reduction in electrolyte.

For the DFT calculations, I am surprised that the monolayer band gap isn't consistently larger than the bulk structures. Can the authors comment.

Ans 7): Thank you for your question.

We have carefully checked all the calculation process and found that there are errors with the parameter setting in the calculation of bulk structure at $x = 0.3$, resulting in the deviation of the calculation results. After correction, we recalculated the data at $x=0.3$ carefully, and we have revised the corresponding content of the manuscript. After computational optimization, the new energy band structure and PDOS are obtained by further calculations. The recalculated bandgap of bulk gersiloxene with $x=0.3$ is 1.66 eV, which is smaller than the monolayer structure and is consistent with the trends of other gersiloxenes.

In supplementary information

Line: 269-274: “The calculated bandgaps for samples with $x=0.1, 0.3, 0.5, 0.7,$ and 0.9 are 1.97, 2.01, 2.15, 2.19, and 2.22 eV, respectively. In addition, bulk structures show the same bandgap type as monolayer structures; that is, they are all direct

bandgap semiconductors, and the bandgap values are 1.58, 1.66, 1.82, 1.96, and 2.05 eV, respectively (Supplementary Fig. 10)”

The authors claim that the material has an internal Type II heterostructure between the Ge and Si, which leads to charge separation. I find that this argument is not well supported since the authors argue with SEM elemental mapping that the 2D layers have an even Si/Ge distribution (no grain boundaries) Therefore, how do you get a type II heterostructure. This statement should be discussed further.

Ans 8): Thank you for your question.

The gersiloxene is an alloy compound of germanium and silicon, which is a single compound rather than a mixture. Therefore, all the as-synthesized gersiloxenes are not really heterostructure. In the part of calculation results analysis, we concluded from the DOS that the distribution of the VBM and CBM indicated that the photoinduced electrons and holes would transfer to the CBM (Si) and VBM (Ge), respectively, which is similar to the trend of electrons and holes transfer in Type II heterojunction. But Ge and Si are in the same molecular structure, and the migration of electrons and holes takes place inside the molecule, so gersiloxene can not be called heterojunction. We mention type II heterostructures only for the convenience of understanding the electronic structure of gersiloxene, as is often done in the theoretical calculations part of literature. (Applied Catalysis B: Environmental, 2017, 217, 275; J. Mater. Chem. A, 2016, 4, 12913; J. Mater. Chem. A, 2013, 1, 2231; RSC Adv., 2016, 6,953; Jpn. J. Appl. Phys., 2019, 58, 030906) Moreover, to avoid confusion, we have revised the introduction part of the manuscript. And we insert the cited literature in Supplementary Information.

Line 94-98: “Among the synthesized gersiloxenes, the distribution of the VBM and CBM for gersiloxene with $x=0.5$ (HGeSiOH), is more conducive to the separation of excited electrons and holes, and it has a wide spectral response range from the ultraviolet to the near infrared region, which is most beneficial for the

photogeneration of electrons and holes.”

In supplementary information

Line 306-310: “That is, for gersiloxene with $x=0.5$, i.e., HGeSiOH, the distribution of the VBM and CBM is similar to that of type-II heterostructures,⁷⁻¹¹ and the photoinduced electrons and holes would transfer to the CBM (Si) and VBM (Ge), respectively, which is more conducive to the separation of excited electrons and holes.”

Reviewers' comments:

Reviewer #1 (Remarks to the Author):

Although the authors have considerably improved the paper, I still cannot accept it as a short communication. The reason for this decision is:

A manifold of semiconductor powders are much easier accessible and photocatalyze reduction of water or carbon dioxide. So, the photocatalytic properties of the novel materials are within expectation and there is no need for a short paper.

Reviewer #2 (Remarks to the Author):

I feel my three points raised in the previous round of review have been satisfactorily addressed. A calculated mistake in band structure for $x=0.3$ has been corrected. The according statements to reply the three points have been added in the manuscript or supplementary information.

Reviewer #3 (Remarks to the Author):

The manuscript by Zhao et al. "Novel two-dimensional gersiloxenes with tunable bandgap for photocatalytic H₂ evolution and CO₂ photoreduction to CO" describes the synthesis and application of siligenes and geriloxenes.

Overall, I am still not convinced that this is ready for publication. The paper should be reconsidered after major edits.

The article suffers from a lack of organization and clarity. With this much data being presented, the manuscript would benefit from more organization using section headers. Also, I would like to see all of the material characterization in one place, rather than scattered throughout the text. Similarly, the SI should be more clearly organized. As it is now, there is very little order to the presentation of Figures and it doesn't follow the text. I would suggest reordering the SI to follow the main text. I would also suggest section headers for the SI.

In Figure 2, I would suggest using different color for the text and arrows in the middle column of figures.

In Figure 3, it is confusing how the x-axis changes from nm to microns. These should be consistent. Also, it is hard to see the image when it overlays with the line trace. I would consider making sure that these do not overlap within the panel.

With regards to the scientific findings, I do think that this is a novel experiment and has a lot of good results. However, this is hard to pull out of the manuscript because of the reasons above.

Figure 4d: In the O1s spectra, I am confused about the Ovs assignment. Is this the O1s assignment for the vacancy? How can the lack of O give intensity?

For this high impact journal, the introduction should be more broadly motivated as to why the community should care about these materials. As it reads now, it does not give the reader context into how important these materials could be for various applications and fundamental studies.

Response to Reviewers' Comments

Reviewer #1 (Remarks to the Author):

Although the authors have considerably improved the paper, I still cannot accept it as a short communication. The reason for this decision is:

A manifold of semiconductor powders are much easier accessible and photocatalyze reduction of water or carbon dioxide. So, the photocatalytic properties of the novel materials are within expectation and there is no need for a short paper.

Ans 1): Thank you for your comments.

It is true that a manifold of semiconductor powders are suitable for photocatalytic reduction of water or carbon dioxide. Among these photoactive semiconductors, two-dimensional (2D) nanomaterials show superior photocatalytic performances due to their extraordinary advantages such as the atomic thickness, larger surface-to-volume ratio, good conductivity, superior electron mobility, and the high fraction of coordinated unsaturated surface sites. (Chem. Commun. 2014, 50, 10768-10777; Chem. Soc. Rev. 2012, 41, 782–796; Nat. Commun. 2012, 3, 1057; Sustainable Energy Fuels, 2017, 1, 1875-1898; ACS Nano 2019, 13, 8566-8576) As a result, 2D semiconductors become one of the hot issues in the photocatalytic fields, and provide a wide range of opportunities for constructing diverse forms of composite photocatalysts with high activity for reduction of water or carbon dioxide. (Chem. Rev. 2017, 117, 6225–6331; Catalysis Science & Technology 2017, 7(3), 545–559; ACS Catal. 2018, 8, 2253–2276).

The important properties that qualifies a 2D crystal for photocatalytic reduction of water or carbon dioxide are the suitable bandgap, band edge levels, optical absorption, and charge carrier mobility. A great challenge in artificial photosynthesis is to develop earth-abundant 2D photoactive semiconductors with these important properties. However, most 2D semiconductors has several structural limitations for photocatalysts, for example, group IV 2D nanomaterials including graphene, silicene,

and germanene are zero band gap materials, which is not sufficient to absorb light to drive water or CO₂ reduction (Nature Nanotechnology 2019, 14, 105–106; Nano Lett. 2012, 12, 113-118; Appl. Phys. Lett. 2010, 97, 163114); the absorption range of g-C₃N₄ is mainly limited in the ultraviolet region (Chem. Rev. 2016, 116, 7159–7329; Journal of Solid State Chemistry 2019, 272, 102-112); the monolayer of transition-metal chalcogenides represented by MoS₂ and WS₂ is direct bandgap semiconductor, while the bilayer, trilayer, and multilayer are indirect semiconductors, which will affect the energy conversion efficiency of light (Phys. Rev. Lett. 2010, 105, 136805; Nano Lett. 2010, 10, 1271–1275; Nature Photonics 2016, 10, 227–238; Nanoscale, 2015, 7, 7402). In addition, most 2D semiconductor photocatalysts need a noble-metal cocatalyst to improve the photocatalytic efficiency. (Adv Mater. 2016; 28, 6197; Applied Catalysis B: Environmental 2019, 246, 12-20; Nanoscale 2019, 11, 23126-23131; Journal of Physics D: Applied Physics 2020, 53, 123001; Advanced Materials 2019, 31: 1804710; Applied Catalysis A General 2016, 517, 91-99; Applied Catalysis B Environmental 2013, 136/137, 89-93; Advanced Materials 2016, 28, 2427-2431.)

In our work, a series of new group IV 2D alloy nanomaterials have been designed and prepared, which have great potential for the siligene and 2D-functional materials domain in photocatalysis. Firstly, the as-synthesized gersiloxenes are direct bandgap independent of the number of layers, and the bandgap is tunable in a wide range of 1.8-2.57 eV. Their light absorption range can reach the visible region or even the near-infrared region, which is advantageous to visible light catalytic applications. From which we can choose one with suitable bandgap and energy levels as high-performance photocatalyst without further modification to enhance the catalytic performance. Secondly, oxygen vacancies in gersiloxenes naturally improves the photocatalytic performance of the photoactive semiconductor without further modification to create vacancies. Thirdly, the excellent photocatalytic performance is due to the unique buckled honeycomb silicon-germanium alloy structure, its Si-OH and Ge-H dangling bonds, as well as its own oxygen vacancy. And no noble-metal cocatalyst was used in the photocatalytic tests. The rate and apparent quantum

efficiency are higher than most currently reported photocatalysts, which is without expectation.

In addition, this is the first application example of a germanium-silicon binary alloy two-dimensional material system in photocatalysis. This kind of structural characteristics is of great significance for the study of group IV binary two-dimensional materials and other two-dimensional materials. It represents a proof-of-concept advance in group IV 2D semiconductors in general to facilitate exciton splitting and to improve light harvesting for advanced applications. Moreover, these 2D nanomaterials provide a wide range of opportunities for constructing diverse forms of composite photocatalysts with high activity for water and CO₂ reduction.

In any case, we would like to thank you for your time.

Reviewer #2 (Remarks to the Author):

I feel my three points raised in the previous round of review have been satisfactorily addressed. A calculated mistake in band structure for $x=0.3$ has been corrected. The according statements to reply the three points have been added in the manuscript or supplementary information.

Ans 1): Thank you, we really appreciate your comments.

Reviewer #3 (Remarks to the Author):

The manuscript by Zhao et al. “Novel two-dimensional gersiloxenes with tunable bandgap for photocatalytic H₂ evolution and CO₂ photoreduction to CO” describes the synthesis and application of siligenes and geriloxenes.

Overall, I am still not convinced that this is ready for publication. The paper should be reconsidered after major edits.

1. The article suffers from a lack of organization and clarity. With this much data being presented, the manuscript would benefit from more organization using section headers. Also, I would like to see all of the material characterization in one place, rather than scattered throughout the text. Similarly, the SI should be more clearly organized. As it is now, there is very little order to the presentation of Figures and it doesn't follow the text. I would suggest reordering the SI to follow the main text. I would also suggest section headers for the SI.

Ans 1): Thanks for the constructive suggestion.

We tried our best to improve the manuscript and made some changes in the manuscript.

First, we have added section headers to the manuscript and the SI.

Section headers:

“Characterization of the resulting materials.”

“Band structure analysis.”

“Photoreduction activity of the resulting materials.”

“Adsorption energies analysis.”

were added to the manuscript.

And we added the following table of contents to the SI.

“Contents

1. Experimental Section

2. Supplementary Figures and Tables

3. References”

section headers:

“Morphology, crystal structure and surface characteristics of the precursor $\text{CaGe}_{2-2x}\text{Si}_{2x}$.”

“Morphology, energy-dispersive spectroscopy (EDS) and specific surface area of the as-synthesized gersiloxenes.”

“Structural models, band structures, VBMs and CBMs of gersiloxenes.”

“VB-XPS, comparison of CO_2 photoreduction under different conditions, and ESR spectra for gersiloxenes.”

“Further discussion on the adsorption energy calculations.”

were added to the “2. Supplementary Figures and Tables” part of SI.

Second, we reordered the manuscript and SI to follow the main text, and here we did not list the changes but marked in blue in revised paper. We also redraw the figures in the manuscript and SI to show all of the material characterization in one place. The following are the redrawn figures.

Fig. 1. (a) Schematic illustration of topotactic deintercalation of $\text{CaGe}_{2-2x}\text{Si}_{2x}$ to

$(\text{GeH})_{1-x}(\text{SiOH})_x$ ($x < 0.5$) or $(\text{GeH})_{1-x}\text{Si}_x(\text{OH})_{0.5}\text{H}_{x-0.5}$ ($x \geq 0.5$) (Ca, blue; Ge, green; H,

white; O, red; Si, yellow). (b) XRD patterns of the topotactic deintercalation products

$\text{Ge}_{1-x}\text{Si}_x\text{H}_{1-y}(\text{OH})_y$ ($x=0, 0.1, 0.3, 0.5, 0.7, 0.9, 1$), GeH and $\text{Si}_6\text{H}_3(\text{OH})_3$.

Fig. 5. (a) UV-vis diffuse reflectance spectra and (b) Tauc plots of gersiloxenes ($x=0.1, 0.3, 0.5, 0.7, 0.9$), GeH and $\text{Si}_6\text{H}_3(\text{OH})_3$. (c) The optical images of GeH, $\text{Si}_6\text{H}_3(\text{OH})_3$, and gersiloxenes with $x=0.1, 0.3, 0.5, 0.7, 0.9$. (d) Energy band structure of gersiloxenes with different x values and GeH and $\text{Si}_6\text{H}_3(\text{OH})_3$ for CO_2 reduction to CO and H_2 evolution.

Fig. 6. Time-dependent photocatalytic hydrogen evolution (a) and HERs (b) of gersiloxenes with $x=0.1-0.9$, GeH and $\text{Si}_6\text{H}_3(\text{OH})_3$. (c) Photostability for H_2 production of the gersiloxene with $x=0.5$ (HGeSiOH). Time-dependent photocatalytic CO evolution (d) and COERs (e) of gersiloxenes with $x=0.1-0.9$, GeH and $\text{Si}_6\text{H}_3(\text{OH})_3$. (f) Time-dependent photocatalytic CO evolution for 10 h. PL

spectra **(g)**, EIS Nyquist plots **(h)** and transient photocurrent responses (300 W xenon lamp) **(i)** of gersiloxenes with $x=0.1-0.9$, GeH and Si₆H₃(OH)₃.

Supplementary Fig. 3 XRD patterns of the precursor CaGe_{2-2x}Si_{2x} (x=0, 0.1, 0.3, 0.5, 0.7, 0.9, 1), CaGe₂ and CaSi₂. The vertical lines at the top and bottom in (a) are the Inorganic Crystal Structure Database (ICSD) for CaSi₂ (ICSD154431) and CaGe₂ (ICSD245612). (b) The corresponding enlarged XRD patterns.

2. In Figure 2, I would suggest using different color for the text and arrows in the middle column of figures.

Ans 2): Thanks for the constructive suggestion.

We have redrawn the figures using a different color for the text and arrows in the middle column of figures.

Fig. 2. (a-e) Low-magnification TEM images, (f-j) HRTEM micrograph, and (k-o) electron diffraction patterns of Ge_{1-x}Si_xH_{1-y}(OH)_y (x=0.1, 0.3, 0.5, 0.7, 0.9) sheets.

(a)(f)(k) x=0.1, (b)(g)(l) x=0.3, (c)(h)(m) x=0.5, (d)(i)(n) x=0.7, (e)(j)(o) x=0.9.

3. In Figure 3, it is confusing how the x-axis changes from nm to microns. These should be consistent. Also, it is hard to see the image when it overlays with the line trace. I would consider making sure that these do not overlap within the panel.

Ans 3): Thanks for the constructive suggestion.

We have redrawn the thickness curves of AFM to make x-axis consistent, and we have also rearranged the AFM images and pictures thickness curves to avoid overlapping.

Fig. 3. AFM images and corresponding height profiles of $\text{Ge}_{1-x}\text{Si}_x\text{H}_{1-y}(\text{OH})_y$

nanosheets. (a-e) $x=0.1, 0.3, 0.5, 0.7, 0.9$.

4. Figure 4d: In the O1s spectra, I am confused about the Ovs assignment. Is this the O1s assignment for the vacancy? How can the lack of O give intensity?

Ans 4): Thank you for the question.

We are very sorry for the confusion, because we didn't explain it clearly.

The O1s peak at 531.7 eV is not directly assigned to the vacancy, but it reflects the concentration of the Ovs. Because Ovs is unstable, when it is produced, it will quickly absorb the oxygen species in the surrounding environment. The adsorbed oxygen can stabilize the oxygen vacancy. Therefore, the concentration of oxygen vacancy is reflected by the intensity of the adsorbed oxygen. The O1s peak at 531.7 eV is assigned to the O atoms near the oxygen vacancy, which reflects the intensity of Ovs. (Chem. Commun., 2016, 52, 5316-5319; Angew. Chem. Int. Ed., 2018, 57, 6054 – 6059; J. Am. Chem. Soc., 2014, 136, 6826–6829; Applied Catalysis B: Environmental, 2018, 239, 68–76; RSC Adv., 2018, 8, 5652-5660; Applied Catalysis B: Environmental, 2018, 221, 187-195; Solar Energy Materials and Solar Cells, 2017, 171, 24–32.)

We have revised the manuscript.

Line 227-234: “Moreover, O1s spectrum in the range of 527 eV~536 eV exhibits the states of surface oxygen on samples (Fig. 4d). The O1s peaks can be also fitted into several components. Peaks at around 531, 532.4, and 533.1 eV are attributed to the Ge-O, Si-O and O-H, respectively. For GeH, Si₆H₃(OH)₃ and all gersiloxenes, a same fitted peak located at around 531.7 eV, corresponded to the O atoms in the vicinity of oxygen vacancies, is supposed to be associated with the surface oxygen vacancies,⁵⁷⁻⁵⁸ which would significantly influence their optical properties.”

5. For this high impact journal, the introduction should be more broadly motivated as to why the community should care about these materials. As it reads now, it does not give the reader context into how important these materials could be for various applications and fundamental studies.

Ans 5): Thank you for the question.

We have revised the introduction to describe the importance of these materials and point out their important potential in fundamental studies as well as various applications.

Line 39-96: “As group IV graphene analogs, silicene and germanene have unique honeycomb structure with low buckling amplitude and sp^2 - sp^3 hybridized bonds that are different from graphene,¹ therefore show many novel physical and chemical properties, and are expected to become graphene's strong competitor. These germanium (Ge) and silicon (Si) based two-dimensional (2D) materials have been widely explored in various applications including electronic devices^{2,3}, photodetectors⁴, chemical sensors⁵, batteries^{6,7}, catalysis^{8,9}, and topological insulators^{10,11}. However, unlike a large number of experimental applications of graphene, most of the studies of germanene and silicene are in the stage of theoretical research, which still needs a lot of experimental data to verify. This is principally because they don't have counterpart bulk materials and can only be chemically synthesized in small amounts by few costly ways such as epitaxial growth^{12,13} and electrochemical synthesis¹⁴. And most importantly, silicene and germanene are the so called zero-gap semiconductors (band gap values close to 0 eV)^{15,16}, which limits their application in nanoelectronic devices, photodetectors, and catalysts with high performance. Opening the bandgap to regulate electronic properties is crucial for exploring their potential applications. Hydrogenation is an effective approach to tailor the electronic properties of silicene and germanene¹⁷⁻¹⁹. The hydrogenated silicene and germanene¹⁹, which are named silicane (SiH) and germanane (GeH), respectively, have been reported in computational analysis and experimental results. Germanane is a direct-gap semiconductor with an experimental bandgap of 1.59 eV (1.56 eV in theory)²⁰, while silicane has an indirect bandgap of 2.3 eV²¹ (2.75 eV²² or 2.94 eV²³ in theory). Moreover, -H/-OH terminal-substituted silicene (siloxane, $Si_6H_3(OH)_3$) has a direct bandgap of 2.4 eV²⁴ (2.5²⁵ or 2.2 eV²⁶ in theory). The opened bandgap by hydrogenation makes it possible for germanene and silicene to be used for photocatalysis²⁷⁻²⁹, which is a highly efficient, low energy consumption, clean, and non-secondary pollution technology for broad application fields involving solar

conversion and storage and contaminants degrading. Achieving regulation of the bandgap can directly control the light absorption capacity of the semiconductor material, thereby improving the photocatalytic performance³⁰.

Alloying is one of the most effective ways to tailoring the bandgap of semiconductor materials. Ge and Si are known to be completely miscible in any ratio, and GeSi random alloys have been investigated for many years³¹⁻³⁵. Therefore, it is possible to form 2D GeSi alloys (siligene, SiGe) with a graphene-like structure similar to that of germanene and silicene. In fact, 2D honeycomb $\text{Si}_{1-x}\text{Ge}_x$ has been demonstrated in theory to be energetically stable because Si and Ge atoms have similar covalent radii, which enable their honeycomb geometry to deform a little to accommodate different atoms^{36,37}. The electronic properties of 2D $\text{Si}_{1-x}\text{Ge}_x$ alloys can be tuned by the value of x ³⁸. Silicene and germanene show slightly buckled configurations with buckling amplitudes of 0.46 Å and 0.68 Å, respectively, whereas siligene is predicted to be stable with a buckling parameter of 0.58 Å³⁷. Moreover, tuning the band gap of siligene by H, F, Cl³⁹ and Br atom modification⁴⁰; alkali metal adsorption⁴¹; and surface functionalization has also been proven to be possible in theory⁴². Xia et al.⁴³ predicted that fully hydrogenated honeycomb $\text{Si}_x\text{Ge}_{1-x}\text{H}$ alloys have finite gaps in the range of 1.09–2.29 eV for x in the whole range from 0 to 1 and that the gap type undergoes a transition between a direct gap and an indirect gap around the composition $x=0.7$. Jamdagni et al.³⁷ confirmed that the chair conformation is energetically favorable for monolayers of pristine and hydrogenated siligene. HSiGe (semihydrogenation; H atoms are merely bonded to Si atoms) exhibits the characteristics of magnetic semiconductors with a bandgap of ~0.6 eV. However, until now, there have been few studies on the preparation of siligene and its derivatives. The topochemical transformation of Zintl-phase CaGe_2 and CaSi_2 into germanane and silicane provided ideas for the preparation of two-dimensional silicene and germanene derivatives, which indicated a possibility for their extensive application. The creation of 2D Ge/Si alloy analogues of germanane and silicane would allow a better understanding of how the electronic structure, optical properties can be tuned to realize enhanced optoelectronic properties and photocatalysis

applications.”

We appreciate for reviewers’ warm work earnestly, and hope that the correction will meet with approval.

Once again, thank you very much for your comments and suggestions.

REVIEWERS' COMMENTS:

Reviewer #3 (Remarks to the Author):

The revised manuscript has addressed most of my concerns. I would urge the the authors to relabel the O1s peak for Ovs in Figure 4f. I would change to Oads and define as adsorbed O at vacancy sites.

Also, I rather like the explanation to Reviewer 1 regarding why these materials are different from other semiconducting powders. I think that this is a stronger introduction than some of the current changes that were made in the last round. I would suggest using part of that response as the introduction.

I would suggest that this would be ready for publication.

Response to Reviewers' Comments

REVIEWERS' COMMENTS:

Reviewer #3 (Remarks to the Author):

The revised manuscript has addressed most of my concerns. I would urge the the authors to relabel the O1s peak for Ovs in Figure 4f. I would change to Oads and define as adsorbed O at vacancy sites.

Also, I rather like the explanation to Reviewer 1 regarding why these materials are different from other semiconducting powders. I think that this is a stronger introduction than some of the current changes that were made in the last round. I would suggest using part of that response as the introduction.

I would suggest that this would be ready for publication.

Q1) The revised manuscript has addressed most of my concerns. I would urge the authors to relabel the O1s peak for Ovs in Figure 4f. I would change to Oads and define as adsorbed O at vacancy sites.

Ans 1): Thanks for the constructive suggestion, and thank you for your comments.

We have relabeled the O1s peak as Oads for Ovs in Figure 4f, and revised the manuscript to define the adsorbed O at vacancy sites.

Line 210-214: “For GeH, Si₆H₃(OH)₃ and all gersiloxenes, a same fitted peak located at around 531.7 eV, corresponded to the adsorbed O atoms in the vicinity of oxygen vacancies (Oads), is supposed to be associated with the surface oxygen vacancies⁵²⁻⁵³, which would significantly influence their optical properties.”

Fig. 4.

Q2) Also, I rather like the explanation to Reviewer 1 regarding why these materials are different from other semiconducting powders. I think that this is a stronger introduction than some of the current changes that were made in the last round. I would suggest using part of that response as the introduction.

Ans 2): Thanks for the constructive suggestion. We have revised the manuscript by using part of the explanation to Reviewer 1 as the introduction.

Line 39-94: “Photocatalysis has attracted wide attention due to the highly efficient, low energy consumption, clean, and non-secondary pollution advantages. 2D nanomaterials provide a wide range of opportunities for constructing diverse forms of composite photocatalysts with high activity for photocatalysis, due to their extraordinary advantages such as the atomic thickness, larger surface-to-volume ratio, good conductivity, superior electron mobility, and the high fraction of coordinated unsaturated surface sites^{1,2}. However, the important properties that qualifies a 2D crystal for photocatalytic application are the suitable bandgap, band edge levels, optical absorption, and charge carrier mobility³. Most 2D materials have several structural limitations for photocatalysts, for example, graphene is zero-bandgap material⁴, which is not sufficient to absorb light to drive photocatalytic oxidation or reduction reaction; the absorption range of g-C₃N₄ is mainly limited in the ultraviolet region²; the monolayer of transition-metal chalcogenides represented by MoS₂ and WS₂ is direct bandgap semiconductor, while the bilayer and multilayer are indirect semiconductors⁵, which will affect the energy conversion efficiency of light. In addition, most 2D semiconductor photocatalysts need a noble-metal cocatalyst to improve the photocatalytic efficiency⁶.

Silicene and germanene are group-IV 2D-Xenes analog to graphene, and are also the so-called zero-bandgap materials but with direct bandgap of 1.55 and 23.9 meV^{7,8}, respectively. They have better tunability of the bandgap than graphene. Therefore, the bandgap engineering of silicene and germanene has been widely explored in various

applications including electronic devices^{9,10}, photodetectors¹¹, chemical sensors¹², batteries^{13,14}, catalysis¹⁵, and topological insulators^{16,17}. It is proved by theoretical and experimental results that silicene, germanene and their derivatives have great potential in photocatalysis¹⁸⁻²⁰, and one of the silicene derivatives has been demonstrated to be a metal-free semiconductor for photocatalytic water splitting²¹.

Hydrogenation and alloying are two effective ways to tailor the bandgap^{22,23}. The hydrogenation of silicene and germanene have been achieved by the topochemical transformation of Zintl-phase CaGe_2 and CaSi_2 into germanane (GeH)²⁴ and silicane (SiH)²⁵. Another hydrogenation product of silicene is siloxene ($\text{Si}_6\text{H}_3(\text{OH})_3$) which has also been synthesized by the similar methods²⁶. 2D honeycomb $\text{Si}_{1-x}\text{Ge}_x$ (siligene) has been demonstrated in theory to be energetically stable because Si and Ge atoms have similar covalent radii, which enable their honeycomb geometry to deform a little to accommodate different atoms^{27,28}. Their electronic properties can be tuned by the value of x ²⁹. The $\text{Si}_x\text{Ge}_{1-x}\text{H}$ ³⁰ alloys are predicted to have finite gaps in the range of 1.09–2.29 eV for $0 \leq x \leq 1$. Ge and Si are known to be completely miscible in any ratio, and GeSi random alloys have been investigated for many years³¹⁻³³. However, the creation of siligenes and their derivatives have rarely been reported. The synthesis of 2D Ge/Si alloy analogues of germanane and silicane would allow a better understanding of how the electronic structure, optical properties can be tuned to realize enhanced optoelectronic properties and photocatalysis applications.

In the study, we report the freestanding siligenes terminated with $-\text{H}/-\text{OH}$ ($\text{Ge}_{1-x}\text{Si}_x\text{H}_{1-y}(\text{OH})_y$, $x = 0.1-0.9$) and name them gersiloxenes, which are synthesized

by the topochemical transformation of freestanding $\text{Ca}(\text{Ge}_{1-x}\text{Si}_x)_2$ alloys prepared by annealing stoichiometric ratios of calcium, germanium and silicon. By combining the experimental results with theoretical calculations, we demonstrate their direct gap type and the bandgap dependence on x (the content of Si), which increase with x values from 1.8 to 2.57 eV. The as-synthesized gersiloxene with $x = 0.5$ (HGeSiOH) is most suitable for photocatalytic hydrogen production and reduction of CO_2 to CO under mild conditions than other gersiloxenes and the germanane and siloxene, due to its moderate band edge levels and bandgap, hybridized orbital composition of the valence band (VB) and conduction band (CB), wide spectral response range, high specific surface area, and oxygen vacancies in gersiloxenes. With no addition of noble-metal cocatalyst, it generates H_2 at a rate of $1.58 \text{ mmol g}^{-1} \text{ h}^{-1}$ in photocatalytic water reduction and CO as the product at a rate of $6.91 \text{ mmol g}^{-1} \text{ h}^{-1}$ in CO_2 photoreduction under mild conditions ($25 \text{ }^\circ\text{C}$, 1 atm CO_2).

We appreciate for reviewers' warm work earnestly, and thank you very much for your comments and suggestions.